# Predicting instabilities in transient landforms and interconnected ecosystems

Taylor Smith ●[1] ✉, Andreas Morr ●[2,3], Bodo Bookhagen ●[1] & Niklas Boers ●[3,4,5]

Many parts of the Earth system are thought to have multiple stable equilibrium states, with the potential for catastrophic shifts between them. Common methods to assess system stability require stationary (trend- and seasonality-free) data, necessitating error-prone data pre-processing. Here, we use Floquet Multipliers to quantify the stability of periodically-forced systems of known periodicity (e.g., annual seasonality) using diverse data without pre-processing. We demonstrate our approach using synthetic time series and spatio-temporal vegetation models, and further investigate two real-world systems: mountain glaciers and the Amazon rainforest. We find that glacier surge onset can be predicted from surface velocity data and that we can recover spatially explicit destabilization patterns in the Amazon. Our method is robust to changing noise levels, such as those caused by merging data from different sensors, and can be applied to quantify the stability of a wide range of spatio-temporal systems, including climate subsystems, ecosystems, and transient landforms.

The study of abrupt transitions has been widely applied to different parts of the Earth system in recent years[1–3], including global vegetation systems[4,5], ice sheets[6], and ocean circulation systems[7,8]. A suite of methods has been proposed to provide early warnings of such transitions, with the most prominent relying on the concept of 'critical slowing down' (CSD) based on slowing dynamics around critical transition points[2,9–11]. Most studies have relied on either a single time series of a key state variable (e.g., paleoclimate proxies[2,10,12]) or sets of time series representing changes in a key variable assessed individually (e.g., vegetation productivity[4,5]). Early warning signals are often measured using the most common CSD indicators—lag-one autocorrelation (AC1) and variance—but can also be explored through other methods such as flickering and skewness[13], via direct, regression-based estimates of the recovery rate as a proxy of stability[4,7], via spectral properties[14,15], or using deep learning techniques[16,17].

The recently introduced dynamic eigenvalue approach[18]—motivated by the estimation of the Jacobian in a time window with locally linear dynamics—directly quantifies stability; if any eigenvalue crosses 1, the system becomes unstable. The application of eigenvalues to the estimation of stability opens the door to new possibilities; in particular, it is possible to move beyond the analysis of a single time series and analyze spatio-temporal grids representing a spatially extended system state through time. In essence, it is possible to study the coherent evolution of an entire system forward in time, to understand whether the system as a whole is destabilizing, and to capture the spatial patterns associated with that (in)stability.

In this work, we first build upon the recently proposed eigenvalue approach to estimate CSD[18] on single time series with a flexible and generalizable methodology based on Dynamic Mode Decomposition (DMD)[19]. We further show how our method can be adapted to natively handle periodicity via the calculation of Floquet Multipliers[20], hence removing a key source of potential bias due to uncertainty in pre-processing methodologies[21]. We illustrate the utility of this method using a glacier system with highly variable seasonal velocities that are difficult to decompose with typical pre-processing methods; in such cases, the application of common CSD indicators (AC1, variance) is not straightforward. Finally, we show how our proposed methodology can also be extended spatially, which opens the door to the study of a

[1]Institute of Geosciences, Universität Potsdam, Potsdam, Germany. [2]Department of Mathematics, School of Computation, Information and Technology, Technical University of Munich, Munich, Germany. [3]Potsdam Institute for Climate Impact Research, Potsdam, Germany. [4]Munich Climate Center and Earth System Modelling Group, Department of Aerospace and Geodesy, TUM School of Engineering and Design, Technical University of Munich, Munich, Germany. [5]Department of Mathematics and Global Systems Institute, University of Exeter, Exeter, UK. ✉e-mail: tasmith@uni-potsdam.de

much wider range of systems and allows for the spatial patterns of destabilization to be analyzed through the tracking of spatial modes associated with instability. We illustrate this approach using synthetic vegetation models, glacier surge dynamics, and the widely discussed potential destabilization of the Amazon rainforest[3,22]. We emphasize that our methodology is flexible to different data types and resolutions, and enables the study of a much larger set of climatic, geomorphic, cryospheric, and environmental processes than have so far been studied under the umbrella of CSD.

## Results

### Estimating the stability of seasonal systems

The core assumption underlying the application of typical CSD approaches is that the system is observed in the form of its stationary fluctuations around a stable fixed point—i.e., that there are no trends or expressions of seasonality that might bias the CSD indicators. In practice, this means that many types of data—for example, vegetation indices—need to be detrended and deseasoned before their stability can be assessed within a CSD framework[21]. While linear detrending is often straightforward, it can become arbitrarily complicated in the case of nonlinear trends such as those imposed on the climate system by slow modes of natural variability; the process of removing seasonality is even more error-prone[21] (Supplementary Fig. S1). A wide range of approaches have been used in the literature, including subtracting long-term climatological means[5], fitting harmonic functions[23], seasonal trend decomposition via Loess (STL[24]), and simple rolling averages over full seasonal periods[21]. While all of these approaches can remove seasonality and yield a nominally stationary residual time series, it is not always clear which parts of the signal should be considered noise, which parts seasonality, and which parts trend. In essence, the process of detrending and deseasoning data can lead to spurious signals which may be misinterpreted as stability changes or over-/under-estimation of resilience[21]. Furthermore, when considering a set of time series (e.g., vegetation indices over a large region), each time series must be deseasoned and detrended individually, potentially introducing additional artifacts due to differences in the removed seasonality of each time series.

However, it is possible to estimate the stability of a dynamical system without enforcing stationarity by analyzing changes in the dynamics superimposed on a regular periodic cycle[20]. One approach is to compute the Monodromy matrix, which is solved over the known or computed periodicity of a given system (Methods). The eigenvalues of that matrix are termed Floquet Multipliers, and quantify stability around a system's periodic orbit[20,25,26]. In practice, this approach provides a framework for quantifying CSD-based resilience or stability in periodic data without first having to remove the periodicity; hence, ambiguity around deseasoning methodologies is removed. In our method, we capture and filter eigenvalues associated with seasonality, and assess whether there remains a non-seasonal eigenvalue that approaches and crosses the stability threshold value of 1 (Methods). We test this assumption using a simple model with seasonality moving towards a critical transition (Fig. 1).

The use of CSD indicators (e.g., AC1, variance, $\lambda$[15]) on non-stationary time series is not supported by the underlying theory of CSD[2,27]; as can be seen in Fig. 1B (red line), CSD indicators are strongly influenced by seasonal fluctuations in the time series, and cannot be used as a warning of a critical transition without first being deseasoned. Floquet Multipliers, however, can capture changes in the underlying driving process around a stable periodicity (Fig. 1C). In short, a stable periodic component and offset are captured by a single strong eigenvalue with a magnitude that remains close to 1, while the process driving the system towards a critical transition is captured by a secondary eigenvalue (or the maximum of all non-periodic eigenvalues). Further eigenvalues capture short-term noise and are relatively low-magnitude (Fig. 1D). The estimation of Floquet Multipliers

(Methods) thus allows for the analysis of seasonal data within a CSD framework without first resorting to deseasoning.

Recent work has identified one further key limitation to using CSD indicators in many cases: multi-instrument data[28,29]. It is often the case that instrumentation upgrades change the fidelity or dynamic range of measurements. These changes—for example, modified signal-to-noise ratios—leave traces especially in the variance of a time series, but also in the autocorrelation[28]. This can make it difficult to differentiate changes in, e.g., the variance driven by a loss of stability and those driven by measurement changes. In this sense, our approach has a key advantage compared to standard CSD indicators: it is less sensitive to changes in noise levels along a time series (Supplementary Fig. S2).

While our method is not immune to noise, the DMD algorithm used here is designed to extract coherent and continuous signals from noisy data; in essence, it attempts to reconstruct the most important parts of the underlying signal despite noise. It accomplishes this through rank truncation (Methods), which is a powerful tool for eliminating short-term and incoherent signals while preserving the main driving process; there also exist several extensions to the basic DMD algorithm designed to handle noise-induced biases[30]. By reconstructing the same data used in Fig. 1 with variable noise levels (Supplementary Figs. S3–S5), we show that noise-level changes within reasonable ranges do not induce strong biases in eigenvalue-based CSD detection (Methods).

### Predicting glacier surging

Recent advances in image cross correlation have yielded global coverage of glacier velocities at high temporal frequency[31,32]. Glaciers are to some degree seasonal—they build mass during the accumulation (e.g., winter) season and lose mass during the ablation (e.g., summer) season. The speed at which glaciers move is roughly proportional to their mass; heavier glaciers tend to move faster[33], given similar slope and bedrock conditions. In many glaciers, downslope movement is not regular, but rather moves in distinct phases. At low mass loadings, glaciers can become 'locked' and move very slowly or not at all. At some critical threshold, many glaciers can start to move very quickly in what is known as a glacier surge. While the exact conditions that trigger such surges are still debated[33–39], many glacier regions worldwide are known to have recurrently surging glaciers.

Unlike vegetation data, which has been studied extensively using CSD[4,5,22], glacier velocity data is ill-suited to deseasoning techniques. The variability in seasonal velocity amplitudes can be several orders of magnitude; typical long-term mean or rolling-window estimates of seasonality, hence, leave large traces in the nominally deseasoned and stationary residual time series as required for using AC1 and variance to quantify stability and its changes within the CSD framework. To overcome this challenge, we estimate Floquet Multipliers across a set of moving windows (Methods) instead of typical AC1 and variance estimates. We use velocity time series at single points over the period 2014-2025 for two test glaciers, one in Alaska (RGI2000-v7.0-G-01-13271, [−140.847, 60.089]) and one in the Karakoram (RGI2000-v7.0-G-13-05693, [71.907, 38.837])[40] (Fig. 2).

Seasonality in both glaciers varies significantly through time (Fig. 2); the high and increasing magnitude of the dominant eigenvalue is a clear sign of significant and growing instability in that periodicity for both glaciers. Beyond changes in the seasonal cycle, however, both glacier examples show further destabilization in the non-seasonal parts of the signal, with a lead time of at least 1 year before surging (Fig. 2C, D).

This hints at two possible ways to use Floquet Multipliers in the context of glacier changes: (1) look for strong and consistent increases in the largest/most strongly periodic eigenvalue to quantify destabilization in the seasonal cycle, and (2) examine whether additional eigenvalues are moving towards 1 to indicate non-periodic instabilities related to potential state changes (i.e., glacier surges). Both

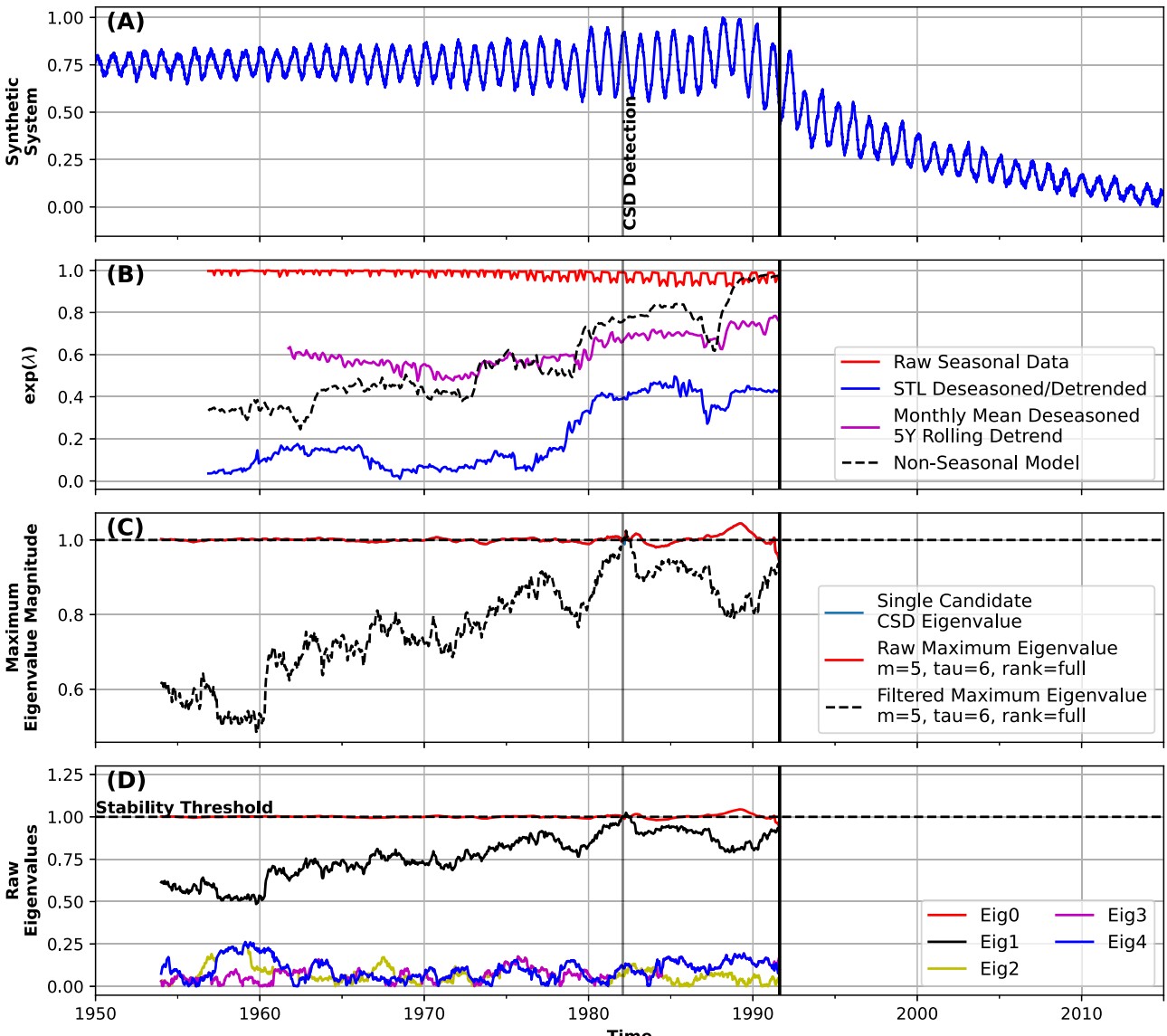

**Fig. 1 | Floquet multipliers. A** Simple pitchfork-bifurcation model with seasonality moving towards a state transition (Methods). Vertical line shows instability detection from eigenvalue tracking. **B** Restoring rate $\lambda$[15] on raw seasonal, non-seasonal, and deseasoned/detrended time series. **C** Filtered eigenvalues, showing that a dominant non-seasonal eigenvalue approaches and crosses 1 before the critical transition. There is also a single eigenvalue that remains close to 1, which captures the stability in the periodicity of the time series (red line). **D** The set of all eigenvalues shows that most of them capture short-term noise in the system rather than coherent changes.

predictions could be of great practical use, even if they can only forecast the onset of glacier surging 1 or 2 years in advance. As glacier surges are key to understanding glacier contributions to sea-level rise[41] and predicting glacier lake outburst flood risk[42], even short warning windows are invaluable.

### Beyond single time series: tracking the stability of the whole system

Single time series are often used to estimate the stability of a system and changes thereof, serving as low-dimensional proxies for the behavior of large, spatially extended systems[7,18]. This simplification is motivated by the ability of a single state variable to reconstruct a wider state-space, often using high-dimensional embedding (e.g., Takens's Theorem[43]). Alternatively, for systems where spatial patterns are of interest, many time series can be analyzed in parallel (e.g., global vegetation resilience maps[4,5]). One of the main motivations for this approach is that deseasoning and detrending are only feasible on individual time series and not regions as a whole; the seasonality of spatially adjacent time series is not guaranteed to be identical.

If we instead consider a spatial area—for example, a single glacier—as a coherent system, we can use spatio-temporal data to understand the development of the system as a whole. Applying the same mathematical framework of DMD eigenvalue tracking and Floquet Multipliers to spatio-temporal snapshots of a given system yields three main advantages: (1) we have a denser and more accurate view of the whole system state; (2) we can visualize which spatial modes contribute to any CSD eigenvalues, and hence understand the spatial patterns of destabilization; and (3) spatially uncorrelated noise will be broadly removed, as it will be expressed as rapidly decaying modes with low eigenvalue magnitudes. We test the utility of this method using a synthetic vegetation model[44], a system with a known transient state (a glacier which surges, cf. Fig. 2), and one that has been proposed to be moving towards a critical threshold, i.e., the Amazon rainforest[3,22].

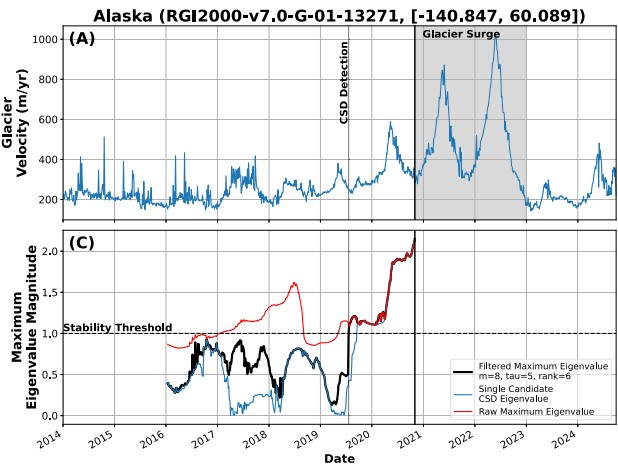

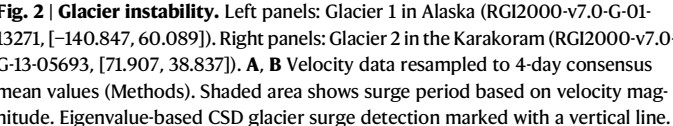

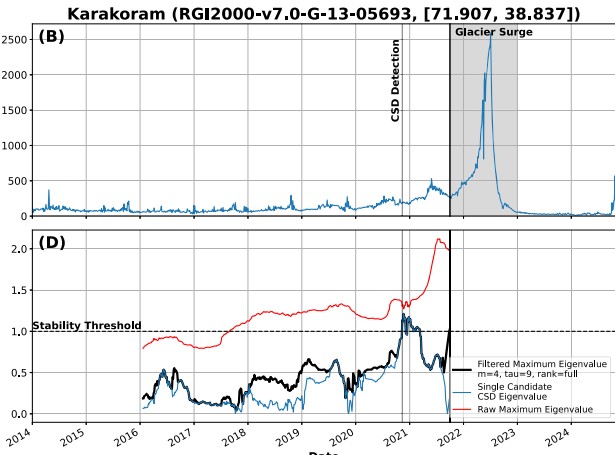

**Fig. 2 | Glacier instability.** Left panels: Glacier 1 in Alaska (RGI2000-v7.0-G-01-13271, [−140.847, 60.089]). Right panels: Glacier 2 in the Karakoram (RGI2000-v7.0-G-13-05693, [71.907, 38.837]). **A, B** Velocity data resampled to 4-day consensus mean values (Methods). Shaded area shows surge period based on velocity magnitude. Eigenvalue-based CSD glacier surge detection marked with a vertical line.

**C, D** Dominant eigenvalues using a period-lagged moving window (period = 1 year). Despite drastic differences in the type of glacier (coastal and mountain) and the relative magnitude of "fast" and "slow" periods, instabilities are both clearly marked by an eigenvalue (or the filtered maximum of non-periodic eigenvalues) crossing the stability threshold (magnitude > 1) before the surge.

**Synthetic vegetation data.** One-dimensional data (e.g., single time series, Figs. 1, 2) is often a drastic simplification of a system under study. From a mathematical perspective, the linear operator that maps a system state forward in time (Methods) can also be estimated from the entire spatio-temporal data cube that describes that system. To test how our Floquet-based analysis works on spatio-temporal data, we adapt a conceptual spatial vegetation model that can simulate transitions from a high- to a low-vegetation state[44] (Fig. 3).

To compare our methodology to established practice, we use a suite of previously introduced spatial early-warning signals (skewness, variance, autocorrelation, Moran's I)[44,45] on a pixel-wise deseasoned and detrended version of the vegetation model (Fig. 3C). We find that temporal autocorrelation and Moran's I increase before the transition, but do not give as early of a warning as with our Floquet-based methodology. Further, the magnitude of the spatial early-warning signals is to some degree biased by the method of deseasoning and detrending the data[21]; calculating spatial early-warning metrics on seasonal data yields strong seasonal fluctuations, precluding their use as early-warning signals (Supplementary Fig. S6).

Using our Floquet-based method, seasonal oscillations in the modeled vegetation are captured in a strong—though slightly unstable—seasonal eigenvalue (Fig. 3D), alongside a clearly increasing non-seasonal eigenvalue crossing 1 from below. We find a definitive warning of an oncoming transition well before the collapse of the vegetation system. It is important to note, however, that defining a single stability threshold for spatially extended systems is challenging; spatial pattern changes will also to some degree influence the computed eigenvalues.

Many real-world systems have additional means by which they adapt to changing environmental stress, and hence avoid state shifts. One clear example of this behavior is spatial pattern formation in vegetation[46–48], where some dryland ecosystems change their spatial organization to adapt to, e.g., water stress. On a simple pattern-formation model[46,48] with seasonal forcing, typical spatial early warning signals do not perform well, as has been previously reported[47]. Our method, however, finds a clear sign of system instability before the onset of strong pattern formation, indicating that Floquet Multipliers are also sensitive to spatial reorganization, not just the approach of critical state transitions (Supplementary Fig. S7). The interpretation of the dominant non-seasonal eigenvalue through that spatial pattern

formation period and onward towards collapse, however, remains unclear. While our first-order analysis is promising, further work on this and other cases where systems exhibit transient coping mechanisms before a state transition would be required to further ground the study of Floquet Multipliers in these cases.

**Glacier surging.** Glacier velocity data can also be considered as spatio-temporal rather than as 1D time series: in essence, the snapshot estimates of velocity over the entire glacier surface can be used instead of velocity measurements taken at a single selected point (Fig. 4). While we use velocities measured along the glacier centerline here for simplicity, gridded velocity estimates over the entire glacier could also be used in the same way; this adds computational complexity but could further refine the spatial patterns of glacier surge initiation. We use 3-year moving windows for computational stability and period-lag by 1 year for the inferred periodicity of glacier movement in order to capture Floquet Multipliers (Fig. 4).

The snapshot spatial field of velocity estimates is already an improved measurement of the system state given that it contains multiple velocity measurements alongside location information rather than just velocity for a single point location. If that pairing (velocity, location) was enough to fully describe how a glacier evolves, it would imply that the stability of the glacier as a whole could be well-predicted from only its current and period-lagged velocity fields. In our tests, this did, in fact, work to predict glacier surging (Supplementary Fig. S8), but adding additional embedding dimensions (i.e., time lags, Methods) gave earlier and stronger warning signals of impending surging; in short, the stability of a glacier is better described by location, velocity, and acceleration over multiple time periods (Fig. 4). In principle, more embedding dimensions can be used; this increases the computational cost and risk of overfitting, but can capture longer wavelength changes in the glacier. In our tests, low embedding dimensions ($m \leq 3$) were sufficient to pick out the signs of an impending surge using the well-monitored period of 2014–2025 (Supplementary Fig. S8), but a higher ($m = 10$) embedding dimension picked out the oncoming surge earlier (Fig. 4). We also tested several different time samplings of the glacier velocity field—between four and 30 days—and found qualitatively similar results (Supplementary Fig. S9).

When we compare the timing of surge initiation in Fig. 4 to that found in Fig. 2C, there is a clear difference—the 1D case seems to spot

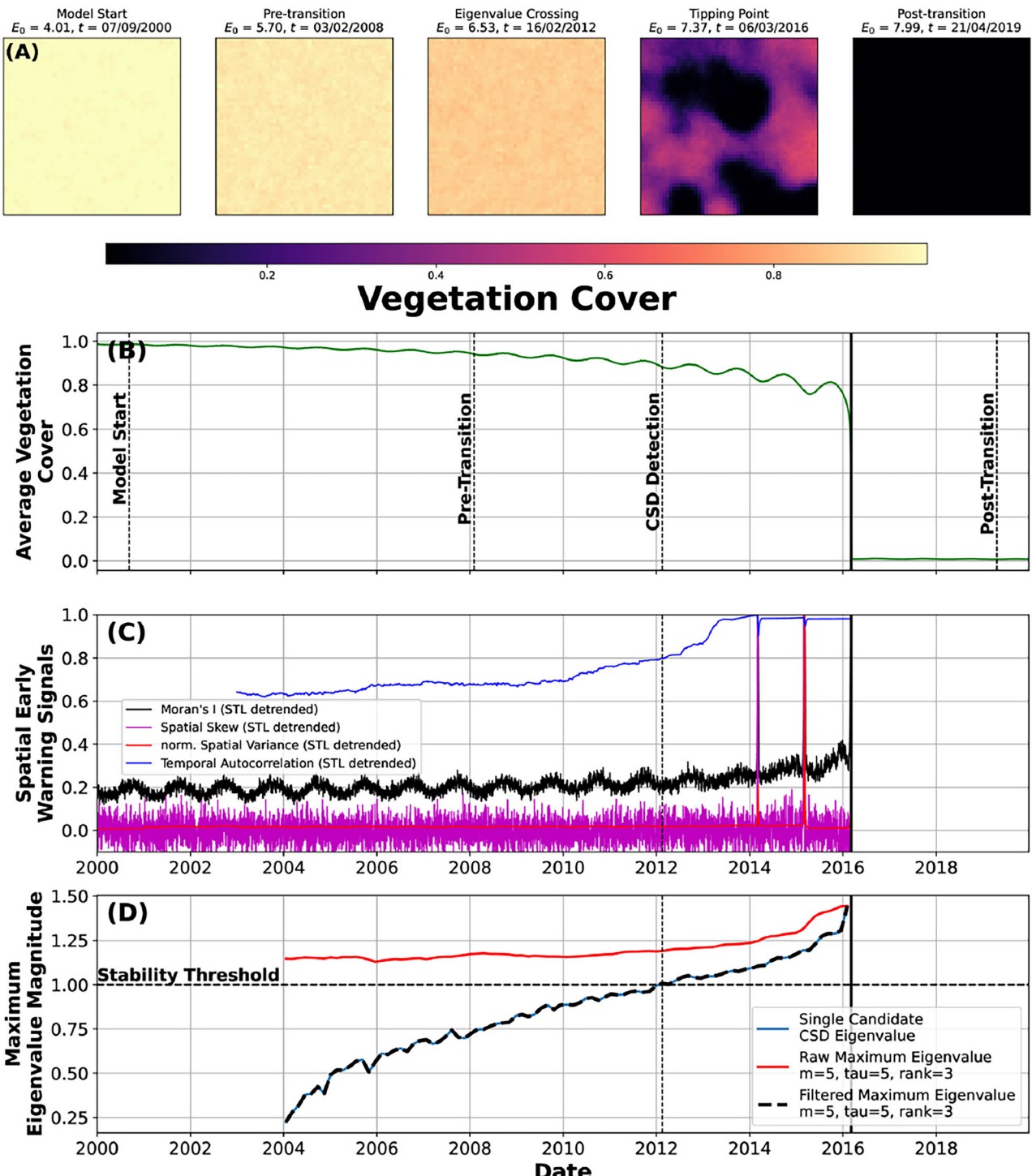

**Fig. 3 | Tipping dynamics from a conceptual spatial vegetation model.**
**A** Snapshot spatial patterns of vegetation state through time, marking different parts of the model run. $E_0$ quantifies environmental stress (Methods). **B** Mean vegetation state, with vertical lines showing the time coordinate of the spatial snapshots from (**A**). **C** Typical spatial early-warning signals estimated on deseasoned and detrended data (Methods). For each time slice: spatial skewness (purple), spatial variance (blue), and Moran's I (black), as well as temporal autocorrelation (red) estimated using a 3-year window (Methods). Temporal Autocorrelation and Moran's I show a rise before the transition; Moran's I, however, also has a clear seasonal oscillation despite being deseasoned. **D** Eigenvalue magnitudes, showing a crossing of the stability threshold well before the transition. The seasonal eigenvalue is slightly unstable (magnitude > 1) due to the slow increase in seasonal amplitude in the model as the resilience of the system decreases; that loss of resilience drives the seasonal amplitude slightly higher through time.

the onset of a surge earlier. It requires, however, that we choose a point on the glacier that is of interest—a choice that is not always easy. By looking at the whole glacier system, we can instead examine the overall stability of the glacier without focusing on a specific point and hence remove possible problems associated with the representativeness of that point. This also allows us to, for example, look at the spatial modes associated with each eigenvalue through time to see which parts of the glacier are contributing to the overall (in)stability of the system (Fig. 4e). From this, we can see that the surge activity is concentrated in distinct areas; it is not expressed as an acceleration evenly throughout

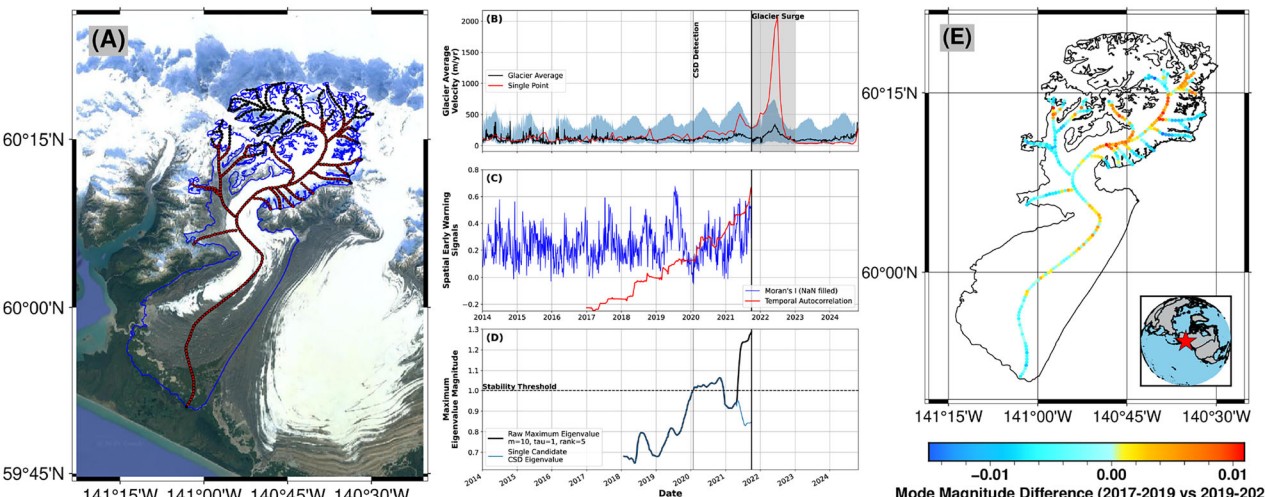

**Fig. 4 | Whole-glacier stability. A** Map of the target glacier in Alaska (RGI2000-v7.0-G-01-13271, [−140.847,60.089]) with outline (blue) and sample points ($n = 337$) along the glacier centerline (red, low-elevation points, black, all centerline points). Background from Google Earth. **B** 4-day ($t = 1004$ time steps) glacier-average velocity (black) plus 25th–75th percentile range (shaded blue), alongside single strongly surging point (red). Shaded area shows surge period based on velocity magnitude. Eigenvalue-based CSD glacier surge detection marked with vertical gray line. **C** Spatial early warning signals on deseasoned/detrended velocity data (Methods). While Moran's I does not show a strong signal, temporal autocorrelation shows a consistent upwards trend. **D** Maximum eigenvalue magnitude using the set of velocities over all low-elevation centerline points, showing a trend towards instability far before the actual surge. **E** Change in the spatial mode (Methods) between stable and directly pre-surge periods. The change indicates that the increase in eigenvalue magnitude is primarily driven by locally activated sections of the glacier. Locator map as inset in bottom right.

the glacier, but rather in spatial subsets that change through time. This illustrates that we capture the growing instability in the glacier despite the spatially disconnected nature of surge initiation.

**The Amazon rainforest.** The ability to track the spatial patterns of destabilization in a coherent system is a key advantage of using spatio-temporal grids instead of 1D time series. To further illustrate this point, we use the widely-studied example of vegetation dynamics in the Amazon rainforest (Fig. 5). A growing body of work has proposed that the Amazon rainforest could potentially tip into a savanna-like state due to a breakdown in the moisture recycling that currently sustains dense vegetation[49]. Previous work by several authors has demonstrated that common CSD metrics—for example, AC1 and variance—indicate that the Amazon has been losing resilience in recent decades[4,22,49]; however, the timing and spatial scale of that critical transition remain unclear.

Using monthly-averaged vegetation optical depth (VOD) estimates[50], we can assess the stability of the Amazon as an entire system, rather than relying on the spatial pattern of individually-analyzed time series (Fig. 5). The spatial mode associated with the eigenvalue of interest—that which is approaching 1 from below—illustrates which part of the system contributes most strongly to that destabilizing process (Fig. 5D); looking at changes in that spatial mode through time also gives insight into which parts of the system are contributing most strongly to the overall destabilization (Fig. 5E). We do not find any strong trend in spatial early-warning signals using deseasoned and detrended vegetation data (Fig. 5B).

The tracked eigenvalue in Fig. 5A has the highest mode magnitude in the southern Amazon (Fig. 5D), which has seen extensive deforestation in recent decades[51] (Supplementary Fig. S10). It is thus not entirely surprising that this area behaves differently from the rest of the Amazon, and that it forms a distinct spatial mode in Amazon vegetation changes (Fig. 5E). Previous studies on resilience changes in the Amazon e.g.,[22,49] have linked loss of resilience to proximity to human activity, low annual precipitation, and a high dependence on recycled moisture. Indeed, Blaschke et al.[49] found a similar spatial pattern of resilience changes (concentration of resilience loss in the

south-west Amazon) using different data and resilience estimation methods. When we compare several different possible drivers of stability loss—deforestation[51], human influence[52], fire frequency[53], precipitation[54], and drought[55]—we find that no single driver stands out as a one-to-one match with the spatial mode changes found in Fig. 5. Some process drivers with a small spatial footprint (e.g., a single road) can have far-reaching ecological consequences; it is thus difficult to make a one-to-one match between changes in the Amazon and stability losses[22].

While the spatial pattern of forest cover loss in the Amazon over the past few decades[51] and a general Human Footprint Index[52] are suggestive, direct coherence of the data sets is low. MODIS Burned Area maps[53] also show relatively higher magnitude in the main southwestern Amazon region where the spatial mode has changed the most, but again do not match cleanly to the spatial pattern of our data. It is thus likely that the implied Amazon destabilization is driven by a range of factors at varying spatial and temporal scales, and further work would be needed to directly attribute stability loss to a suite of drivers.

## Discussion

There is a growing interest across scientific disciplines in incorporating dynamical systems frameworks and especially the CSD concept into research concerned with stability changes in natural systems[3,56–59]. At the same time, several recent publications have underlined critical biases in the most commonly used CSD-based stability indicators—variance and AC1[7,18,21,28,29]; extending the study of system stability to use the well-motivated eigenvalue magnitude approach[18] is a welcome addition. By expanding eigenvalue-based methods of resilience estimation to new domains—both seasonal and spatio-temporal data—the number and diversity of systems that can be studied within a CSD framework increases dramatically.

Eigenvalue-based methods, however, are not without limitations; we identify two key difficulties in assessing uncertainty in our estimated stability changes. First, to capture changes in a system using eigenvalue-based approaches, we need a sufficiently dense picture of

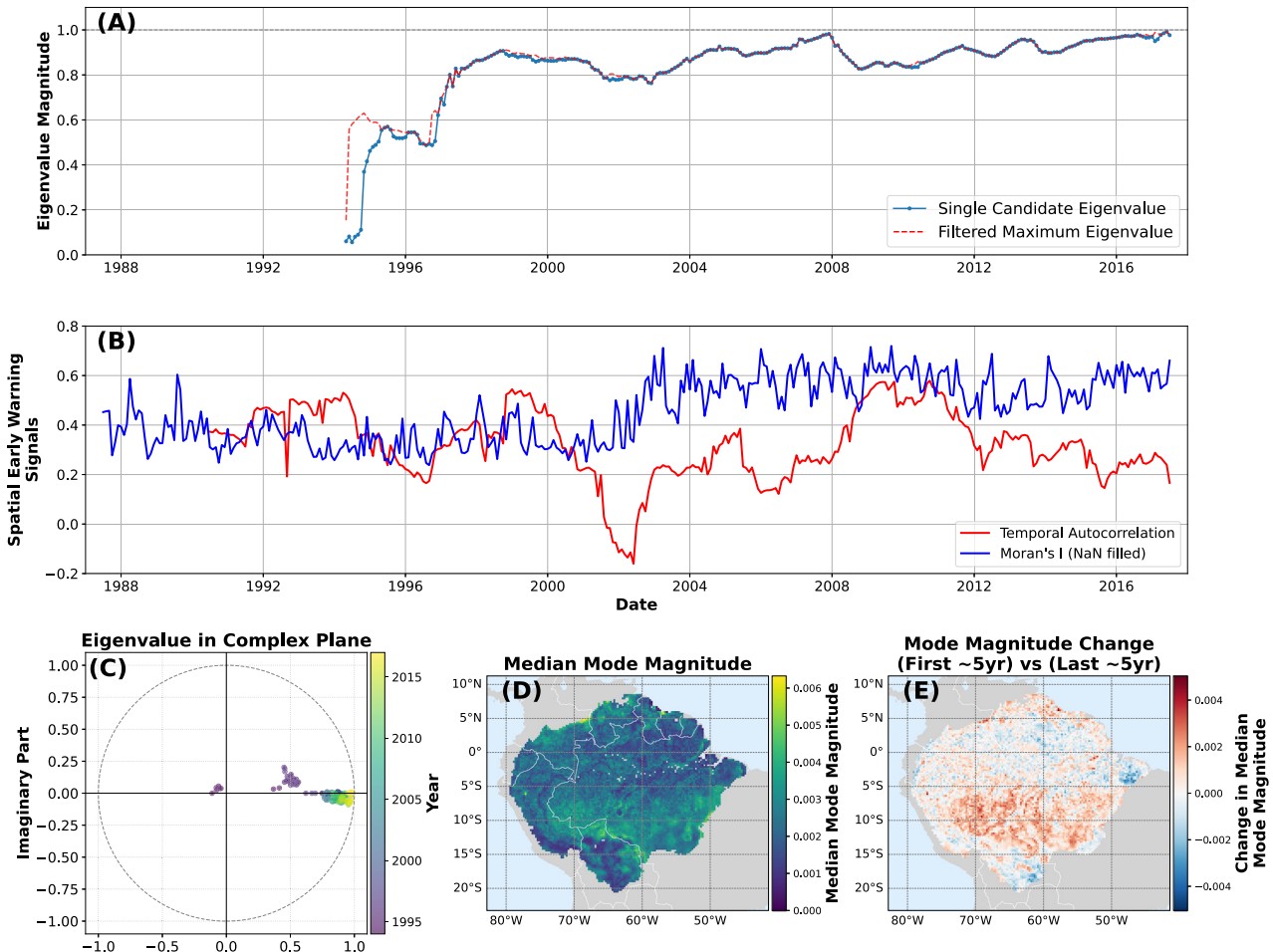

**Fig. 5 | Amazon vegetation stability. A** Single eigenvalue of interest and maximum of all non-seasonal eigenvalues. **B** Spatial early-warning signals (Methods) on deseseasoned and detrended vegetation data, showing no clear temporal pattern. **C** Eigenvalue track in the complex plane showing a movement towards the unit circle over time, indicating loss of stability. **D** Map of the median spatial mode associated with the eigenvalue track of interest, showing that the mode is most strongly associated with the southern parts of the Amazon basin, which have been significantly affected by deforestation in recent decades[51]. **E** Change in the spatial mode map between the first and last 5 years of data. The change indicates that the increase in eigenvalue magnitude is strongest in the southern Amazon, as well as in some areas on the northern edge of the Amazon.

the system state; this is generally accomplished using state-space embedding (Methods)[43,60]. The choice of embedding parameters is not always straightforward—especially for noisy real-world data—and the choice of parameters can influence the implied changes in system stability. Further work is needed to better understand how uncertainty in state-space embedding propagates forward into stability estimates. Second, common bootstrap uncertainty estimates (e.g., using phase surrogates[4]) are not well-suited to our approach, as the phase is intrinsic to how the periodic attractor of interest is defined. There is limited research on bootstrap methods in embedded state-space, and a rigorous analysis of which surrogate-generation methods e.g.,[61–63] are mathematically correct for our method—especially in the context of spatio-temporal data—is outside of the scope of our study.

A further limitation to the use of eigenvalue tracking and Floquet Multipliers in the study of system stability is the choice and availability of data. The example of glacier surging shown here is one with a clear transient state (surging), which is expressed directly and completely in the state variable measured (velocity), which makes for a relatively easy system to study with our method. Landslides behave in some similar ways to glaciers; mass loading increases gravitational downhill forces, stabilized by the cohesion of the soil and friction with the underlying bedrock. The triggering mechanism for a landslide, however, is not as simple as that for a glacier surge; external factors such as rainfall,

geologic rock preconditioning, water table levels, earthquakes, and land-cover changes also play important roles. It is thus not clear if the data available—GPS monitoring, radar interferometry, pixel-tracking velocity fields—is really capturing the system state well enough to predict a critical shift in stability. This same data limitation applies to other Earth system components that could be considered as unstable or multi-state dynamical systems: for example, the earthquake cycle, ranging from stress accumulation to release on a fault line, and the volcanic magma eruption cycle, including magma formation, accumulation, and eruption. The directness of the measurement data—and complications due to constant adjustments to small earthquakes—make the direct application of our analysis framework to those systems difficult.

Remote sensing, however, provides a powerful tool that is perfectly suited to the use of DMD for stability monitoring; indeed, DMD was initially developed for spatio-temporal grids[19,64,65], and has been used to discover dynamical patterns in climate systems[66]. We demonstrate how DMD can be used with remote sensing data through our analysis of glacier (Figs. 2, 4) and Amazon rainforest (Fig. 5) stability. This same framework could be used to study, e.g., ocean circulation systems[7], river avulsion[67], changes in ice export in the north Atlantic[68], or spatial vegetation pattern formation[47]. We emphasize that the mathematical framework we present here is domain-agnostic and can be applied equally well to 1D, 2D, or higher-dimensional data.

Our work builds upon recent theoretical advances in resilience estimation with eigenvalue approaches[18] to present a method that inherently handles seasonality and can also be natively extended to high-dimensional data and spatio-temporal fields. We present examples from synthetic, cryospheric, and ecological systems that are known or thought to have multiple stable or transient states, demonstrating the flexibility and usefulness of our method. It is well-suited to diverse application areas, and it takes advantage of the growing amount of multi-scale data available across diverse spatial and temporal resolutions.

## Methods

### Eigenvalue estimates of linear stability

In many systems, the transition between stable states is preceded by weakening internal feedbacks and stabilizing forces; the slowing of the system's dynamics around critical transition points is broadly referred to as Critical Slowing Down (CSD). The strength of a system's restoring force is often estimated using autocorrelation or variance; there exist, however, alternative means to estimate the strength of that restoring force, and hence the stability or resilience of a given system.

Recent work[18,69] has proposed the use of dynamic eigenvalues as an early warning of critical transitions. This use is motivated by locally linearizing a system around a stable point and calculating the Jacobian. For a stable system, all discrete-time eigenvalues of the Jacobian should be real and less than 1, indicating that perturbations decay back towards the current state. When any eigenvalue approaches or crosses 1 from below, perturbations grow and push the system away from its current state towards a new stable equilibrium. There are many approaches to estimating the Jacobian of a local linearization; for example, Grziwotz et al.[18] use an S-Map[70,71] approach based on a local nearest-neighbors search at every point in a time series. By moving windows across a time series and averaging the dominant (largest) eigenvalue over each window, the stability of the system (magnitude of the eigenvalues) can be estimated through time.

Dynamic Mode Decomposition (DMD) provides an alternative approach to estimating the Jacobian[19,64,65], and hence the stability of a system. In short, DMD finds a set of eigenvalues for a global fit to the system state in one (temporal) window, while S-Map as applied by ref. 18 finds point-wise local eigenvalues and then averages them over a given (temporal) window. For a simple dynamical system with a bifurcation, the two methods yield similar results (Supplementary Fig. S11). This can also be demonstrated with real-world data that does not have such simple dynamics (Supplementary Fig. S12), but was previously shown to be approaching a tipping point[6].

Many natural systems have an inherent periodicity, most often due to daily or annual cycles related to sunlight or temperature. The analysis of such systems within the CSD framework has so far focused on using deseasoned and detrended data, as many commonly used means of estimating stability (e.g., variance, autocorrelation) assume stationary data. Pre-processing data to remove seasonality can introduce biases in resilience estimates, and these biases can vary with the choice of pre-processing methodology e.g.,[21,23,28] (Supplementary Fig. S1). Floquet Multipliers[20,25,26] describe the linear stability of a system over a given period—i.e., in relation to a known periodic forcing such as annual seasonality—and can hence be used to natively quantify the stability of periodic data without resorting to pre-processing. In short, for non-seasonal DMD we use moving windows to look at changes in local stability; for seasonally-forced data, we period-lag our data windows to find the stability of the system around a periodic orbit.

The estimation of system stability via DMD is not limited to single time series; a very wide body of work uses DMD to analyze spatio-temporal data[19,64–66,72]. In our case, the underlying motivation does not change between the single time series and spatio-temporal data cases —we are still estimating the Jacobian of a linear operator that moves the system state forward in time. The change is that we are no longer looking at a single state variable that captures the dynamics of the system, but rather focus on a collection of state variables that change through time (e.g., a spatio-temporal field of data). In both cases, we compute a linear operator and capture its eigenvalues; eigenvalues are a general property of matrices and are hence not limited to a specific $n$D case. While we use spatial fields of single parameters (e.g., velocity), there is no strong reason why multi-dimensional data (e.g., multiple satellite bands) could not also be used in the same way. Furthermore, the underlying theory of using period-lagged data for the estimation of Floquet Multipliers does not change for spatio-temporal data; our DMD approach is thus flexible in that it is applicable to both periodic and non-periodic data which has multiple measurement dimensions (e.g., GPS displacements in x/y/z directions).

**Mathematical framing.** DMD takes snapshot matrices representing subsequent system states and uses singular value decomposition (SVD) to compute a single matrix $\hat{\mathbf{A}}$ that provides the mapping $\mathbf{x}_{k+1} \approx \hat{\mathbf{A}}\mathbf{x}_k$. The main advantage over conventional linear regression techniques is that SVD allows for a prior selection of relevant dynamical modes. This constitutes a reduction of the system dynamics to those that are most important and leads to a more well-posed linear regression problem. The eigenvalues of the linear operator $\hat{\mathbf{A}}$ are a data-driven approximation of the underlying driving process of the system, and provide a means of estimating the stability of the system. For periodically-forced systems, it is possible to estimate changes in linear stability over a given periodicity and hence, estimate resilience within a CSD framework without having to first remove seasonality.

Consider a linear system with a periodic coefficient matrix $\mathbf{M}(t)$:

$$\frac{d\mathbf{z}}{dt} = \mathbf{M}(t)\mathbf{z}, \text{ where } \mathbf{M}(t+T) = \mathbf{M}(t) \tag{1}$$

where $T$ is the known period of the system and $\mathbf{z}$ are perturbations around the periodicity. The fundamental matrix solution is $\mathbf{\Phi}(t, t_0)$, with the state of the system evolving as $\mathbf{z}(t) = \mathbf{\Phi}(t, t_0)\mathbf{z}(t_0)$. The Monodromy matrix $\mathbf{A}$ captures how the system evolves over one full period $T$ as $\mathbf{A} = \mathbf{\Phi}(t_0 + T, t_0)$. The eigenvalues $\{\rho_i\}$ of $\mathbf{A}$ are the Floquet Multipliers[20,25,26], which describe the linear stability of the system over the period $T$. For stable systems, $|\rho_i|$ will be less than 1 for each eigenvalue $\{\rho_i\}$; any eigenvalue >1 implies destabilization in the system.

To compute Floquet Multipliers with DMD, we rely on the link between DMD and Koopman theory[73,74]. Koopman theory allows us to reformulate a non-linear, finite-dimensional system as a linear, infinite-dimensional system, where the infinite-dimensional space is spanned by so-called observables $\phi$. The dynamics of each observable of the system can thus be written as:

$$\phi(t_{k+1}) = \mathcal{K}^T(\phi(t_k)) \tag{2}$$

where $\mathcal{K}^T$ is the linear Koopman operator over the period $T$[73,74]. For any state in the system near the periodic orbit, $\mathcal{K}^T$ is an approximation of the Monodromy matrix $\mathbf{A}$; if the initial system was linear to begin with, these operators are identical. In practice, for snapshots separated by period $T$, DMD can be used to find a finite-dimensional approximation of $\mathcal{K}^T$[74]. This is achieved by computing a linear operator $\hat{\mathbf{A}}$ in the original basis so that $\mathbf{x}_{k+1} \approx \hat{\mathbf{A}}\mathbf{x}_k$, where $\mathbf{x}_{k+1}$ is the system state exactly one period $T$ after $\mathbf{x}_k$. Thus, the $\hat{\mathbf{A}}$ computed with DMD is a data-driven estimate of the linear evolution of the system for one full period $T$. For periodic dynamics, this map is a data-driven approximation of the Monodromy matrix $\mathbf{A}$; hence the eigenvalues of $\hat{\mathbf{A}}$ are approximations of the Floquet Multipliers $\{\rho_i\}$. Thus, we can estimate the stability of the underlying system without having to explicitly remove seasonality, and track eigenvalue changes[18] to understand the stability of inherently periodic systems (Fig. 1).

**State-space embedding.** A core requirement for eigenvalue-tracking-based CSD analysis is that the local linear maps (estimated via S-Map, DMD, or other methods) are computed over a sufficiently dense picture of the underlying system. In practical terms, this means that time series data needs both temporal lags $\tau$ and embedding dimensions $m$ to fully capture the attractor[43,60]. In the examples shown here, $\tau$ is first estimated via testing for minimal information loss across time lags (average mutual information) as $\tau$ is varied from 1 to $T/2$; this helps to diminish the influence of closely co-correlated neighboring time points on the reconstruction of the attractor. Using the discovered $\tau$, we then test how well different embedding dimensions $m$ perform by testing the 'closeness' of points across different embedding dimensions[75], which is a means of estimating how many embedding dimensions are needed to fully reconstruct the attractor. The choice of $m$ and $\tau$ in noisy systems can be difficult; estimates based on the commonly-used Takens's theorem are meant to reconstruct the attractor over infinite and noise-free data, and can fail to find the optimal $m$, $\tau$ parameters in real noisy systems[43,76]. For our synthetic systems, we assess a range of $\tau$ and $m$ parameters, choosing the pair with the lowest $m$ that is able to separate the CSD signal mostly into its own eigenvalue. For the simple case without seasonality, we use a straightforward $m = 1$ setup. The resulting eigenvalue changes found by S-Map and DMD are very similar (Supplementary Fig. S11).

The choice of embedding dimension $m$ and lag $\tau$ for seasonal data via Floquet analysis (Fig. 1) is more difficult due to the confounding influence of the seasonal eigenvalue (close to 1) on the detection of the CSD eigenvalue (below but approaching 1). Delay embedding tests are often not designed to find period-lagged dynamics; for our synthetic seasonal models, they generally reconstruct the strong seasonal attractor and, to some degree, ignore the underlying CSD signal of interest. For each system, we test a range of $m$, $\tau$ values, choosing a pair that cleanly separates out the CSD eigenvalue of interest; the choice of $m$, $\tau$ will depend heavily on the system under study, the driving noise, and the memory time scale of the system.

Once a set of $m$, $\tau$ parameters has been chosen, as well as the SVD rank truncation $r$ and period $T$, the data is processed as a set of overlapping windows. For each window, a set of delay-embedded matrices is formed such that, in the one-dimensional case, a single time point is an $m$-dimensional vector:

$$\mathbf{x}_t = \begin{pmatrix} x_t \\ x_{t+\tau} \\ x_{t+2\tau} \\ \vdots \\ x_{t+(m-1)\tau} \end{pmatrix}$$

where $x_t$ is the value of the time series at time $t$. A single window of data $\mathbf{X}$ of length $n$ is an $m \times n$ matrix:

$$\mathbf{X} = \begin{pmatrix} x_t & x_{t+1} & x_{t+2} & \cdots & x_{t+n-1} \\ x_{t+\tau} & x_{t+1+\tau} & x_{t+2+\tau} & \cdots & x_{t+n-1+\tau} \\ x_{t+2\tau} & x_{t+1+2\tau} & x_{t+2+2\tau} & \cdots & x_{t+n-1+2\tau} \\ \vdots & \vdots & \vdots & \ddots & \vdots \\ x_{t+(m-1)\tau} & x_{t+1+(m-1)\tau} & x_{t+2+(m-1)\tau} & \cdots & x_{t+n-1+(m-1)\tau} \end{pmatrix}$$

This is compared to a period-lagged matrix $\mathbf{Y}$:

$$\mathbf{Y} = \begin{pmatrix} x_{t+T} & x_{t+T+1} & x_{t+T+2} & \cdots & x_{t+T+n-1} \\ x_{t+T+\tau} & x_{t+T+1+\tau} & x_{t+T+2+\tau} & \cdots & x_{t+T+n-1+\tau} \\ x_{t+T+2\tau} & x_{t+T+1+2\tau} & x_{t+T+2+2\tau} & \cdots & x_{t+T+n-1+2\tau} \\ \vdots & \vdots & \vdots & \ddots & \vdots \\ x_{t+T+(m-1)\tau} & x_{t+T+1+(m-1)\tau} & x_{t+T+2+(m-1)\tau} & \cdots & x_{t+T+n-1+(m-1)\tau} \end{pmatrix}$$

The linear operator $\mathbf{A}$ is found via $\mathbf{A} = \mathbf{Y}\mathbf{X}^\dagger$, where $\mathbf{X}^\dagger$ is the Moore-Penrose pseudo-inverse of $\mathbf{X}$. DMD computes this via SVD with optional rank truncation $r$, yielding either the full operator $\mathbf{A}$ or a reduced operator $\tilde{\mathbf{A}}$. In this specific period-lagged setup, the eigenvalues of $\mathbf{A}$ or $\tilde{\mathbf{A}}$ are interpreted as Floquet Multipliers.

**Practical considerations.** The tracking of stability with Floquet Multipliers is complicated by the fact that an eigenvalue indicative of CSD will approach 1 from below, while a stable periodic oscillation (or mean state) will have an eigenvalue of approximately 1[77]. In systems with noise, it can become very difficult to separate eigenvalues representing drift towards a critical transition from those that are associated with stable periodicity. A simple maximum eigenvalue (as used by Grziwotz et al.[18] with S-Map) will capture small changes in the periodicity—not the destabilization of the underlying process. The stable periodic eigenvalue will have, in a perfect case, a value of $[1 + 0j]$; with noise, this can oscillate around 1 and also pick up small imaginary components.

In general, the periodic eigenvalue can be filtered out by removing eigenvalues that are stable and oscillate around 1 throughout the entire period of study, leaving a filtered maximum eigenvalue representing stability change; alternatively, a single candidate eigenvalue can be identified that approaches 1 from below. It is also important to note that mean-centering the windowed data can diminish the magnitude of the stable periodic eigenvalue (e.g., if the amplitude of the periodicity is small), and hence enhance the relative strength of potential CSD modes. We do not mean-center the data in Fig. 1 to illustrate that our method can capture both periodic and non-periodic signals; the impacts of mean-centering can be seen in Supplementary Fig. S13.

For spatio-temporal systems, there are data processing constraints that need to be taken into account, necessitating a smaller SVD rank truncation; mean-centering the data also helps to isolate the eigenvalue of interest when considering complex spatial fields. The maximum number of eigenvalues computed by DMD is limited by either the spatial field ($n$ features) or time ($t$ snapshots) by the minimum of $n - 1$ and $t$ (the 1D case uses $m$ embedding dimensions instead of $n$ spatial locations). Depending on the system, this can generate a massive number of potential eigenvalues, many of which are short-time and decaying features or spatially uncorrelated noise. For example, strong noise in a single time series in a spatial field will not show up coherently throughout all time series, and would be unlikely to show up as a dominant eigenvalue. To rather focus on the main features of interest, we truncate DMD over each moving window and only preserve the dominant eigenvalue(s), letting quickly decaying modes fall away. For each spatio-temporal system studied here (Figs. 3–5), we test a range of $m$, $\tau$, and $r$ rank parameters, with a focus on cleanly separating periodic dynamics from non-periodic dynamics.

In the context of spatio-temporal fields, the structure of the dynamic modes (eigenvectors) associated with any given eigenvalue can be used to further investigate changes in the system. The spatial distribution of the dynamic mode magnitude—especially that associated with a CSD-like eigenvalue—can provide insight into which specific regions contribute most strongly to destabilization and how their spatial pattern has evolved through time (Figs. 4, 5). The dynamic modes are a natural output of DMD alongside the eigenvalues used to track stability here; when time-delay embedding is employed, further processing is needed to extract a single spatial mode from the embedded vector. To investigate only the influence of the current system state on the dynamics, we project the embedded modes back onto their first spatial component vector—only the first $x$-vector of the $m$-embedded spatial $x$-vectors is analyzed. In short, delay embedding can be used to better reconstruct the state space and capture changes in the system; the spatial mode patterns are then removed from that state-space embedding to provide a single spatial mode per time window and eigenvalue.

## Alternative early warning signals of critical transitions

To compare our results to past work, we use a robust method of detecting CSD based on multiple moments of time series statistics that is designed specifically for application in systems with time-correlated noise disturbances which occur generically in natural systems[15,78]. This method—like those based on AC1 and variance—requires the prior mean-centering and deseasoning of time series data. Without knowledge of the exact seasonal signal, it is clear that remnants of the seasonal fluctuations will bias early-warning indicators (Supplementary Fig. S1); to compare a typical workflow for these methods to our results, we deseason and detrend the synthetic data presented in Fig. 1 and Supplementary Figs. S13–S15 using (1) Seasonal Trend Decomposition via Loess (STL[24]) and (2) by removing a long-term monthly mean and a 5-year rolling mean. We find that both of these methods can remove seasonality and trend, but both introduce biases into the resulting estimates of CSD indicators[21] (Fig. 1, Supplementary Figs. S13–S15).

For the spatial cases (synthetic and real-world), we also compare our Floquet analysis to (1) spatial variance, (2) spatial skewness, (3) temporal autocorrelation[45], and (4) Moran's I[44]. These various methods have been used previously on non-seasonal data to discover critical transitions in spatio-temporal systems; they remain, however, influenced by seasonal fluctuations (Supplementary Fig. S6). We deseason and detrend our synthetic and real-world spatial data using STL pixel-wise (Figs. 3–5, Supplementary Figs. S7–S9). We note that the simple grid-wide approach of subtracting a long-term linear fit and an average seasonal amplitude across all pixels individually yields similar results—any deseasoning/detrending method can potentially introduce additional bias in the spatial indicators due to, e.g., different implied seasonalities in neighboring pixels. For conciseness, we only include temporal autocorrelation and Moran's I for our real-world data (Figs. 4, 5).

## Synthetic data

We use five synthetic models (three one-dimensional time series and two spatio-temporal vegetation models) to test our method. Full parameter tables can be found in the Supplement, and code to reproduce each of the models can be found in our code repository[79].

**Single time series tipping models.** To compare standard (e.g., AC1, $\lambda$[15]) and eigenvalue-based (S-Map and DMD) methods for estimating stability changes in terms of CSD, we use two different synthetic models with bifurcations inducing critical transitions (Fig. 1, Supplementary Figs. S3–S5, S11, S13–S15).

We generate a seasonally-forced pitchfork model:

$$\frac{dx}{dt} = p(t)\, x - x^3 - \lambda x + A\cos(\omega t) + \sigma\,\xi(t) \qquad (3)$$

where the control parameter $p$ is varied linearly through time to generate the transition (Fig. 1). We set the linear damping coefficient $\lambda = 0$ so that the system is driven by periodic forcing of amplitude $A$ and periodicity $\cos(\omega t)$, with additive Gaussian white noise $\sigma\xi(t)$. We use the same model where the control parameter $p$ is varied sigmoidally, showing that even for rapid tipping we still generate useful early-warning of the oncoming transition (Supplementary Fig. S14).

We also generate a similar seasonally-forced logistic model:

$$\frac{dx}{dt} = r(t)\, x\left(1 - \frac{x}{K}\right) + A\cos(\omega t) + \sigma\xi(t) \qquad (4)$$

with a time-varying control parameter $r$ and a carrying capacity $K = 1$, as well as additive Gaussian white noise $\sigma\xi(t)$ and periodic forcing $A\cos(\omega t)$ with an annual periodicity. It is important to note that this model is clipped to $x \geq 0$ during the integration. Both models use Euler-

Maruyama for numerical integration. We produce two versions of each model—with and without seasonality—by setting the amplitude $A$ to zero; this provides a non-seasonal reference for the same tipping system. We use the pitchfork model with seasonality for Fig. 1 and Supplementary Figs. S3–S5, S13, S14, and without seasonality for Supplementary Fig. S11. We use the logistic model for Supplementary Fig. S15.

**Stationary process model.** To test the impact of variable noise levels on DMD, as would for example arise by merging signals from different sensors[28,29,50], we first generate a stationary process with additive seasonality, transform that time series into slices of variable length with different additional additive noise, and compute a daily average (Supplementary Fig. S2). To generate a stationary residual for AC1 and variance calculation, we use STL[24]; we do not pre-process the daily signal for eigenvalue tracking. As a second experiment, we add time-varying and additive noise to the same pitchfork model used in Fig. 1. We (1) decrease, (2) increase, and (3) shuffle the noise levels through time (Supplementary Figs. S3–S5), in each case capturing changes in AC1 and variance on STL-processed residuals as well as eigenvalue changes on the daily mean data without pre-processing. We are able to successfully capture the signal of interest (eigenvalue approaching 1 from below) for the decreasing and shuffled noise cases. In the case of very high noise during the transition (noise magnitude >2 times the signal magnitude), the relatively weak CSD mode approaching 1 from below is hard to differentiate from noise, and can thus be missed. We emphasize, however, that this requires very high noise levels; in more typical real-world cases with rather smaller noise magnitudes and changes therein, our approach should remain robust. It is also important to point out that our method does not produce a spurious early-warning signal (as can happen with AC1 and variance), but rather does not find a CSD-like eigenvalue for very high noise levels.

**Spatio-temporal vegetation models.** To expand our analysis to spatio-temporal data, we rely on two additional tipping models: (1) a simple reaction-diffusion model modified from Dakos et al.[44] to have a seasonal component, and (2) a spatially extended Klausmeier pattern-formation model, also modified to add a seasonal component[46,48,80].

The reaction-diffusion model is parameterized for vegetation $V$ through time as:

$$\frac{\partial V}{\partial t} = r_v(x, y, t)\, V\left[1 - V\frac{(h_E^p + E^p)}{h_E^p}\right] + D\nabla^2 V + \sigma\,\eta(x, y, t) \qquad (5)$$

with vegetation growth rate $r_v$, environmental stress $E$, diffusion coefficient $D$, noise amplitude $\sigma$. The noise $\eta(x, y, t)$ is spatially and temporally uncorrelated Gaussian noise. The parameters $h_E$, $h_v$, and $p$ are used to control the rate and shape of the critical transition by modifying the stressor $E$. The stressor $E$ is related to vegetation cover as:

$$E = E_0(t)\, \frac{h_v}{h_v + V} \qquad (6)$$

To make the model seasonal, we add a seasonal component to both $r_v$ and $E$. For $E$, we maintain the same annual-scale, homogeneous, seasonal forcing across all spatial cells with amplitude $A_{E_0}$ as the stressor is slowly pushed across a transition:

$$E_0(t) = E_{0,\,\text{base}}(t) + A_{E_0}\sin\left(\frac{2\pi t}{T_{\text{year}}}\right) \qquad (7)$$

To prevent phase-locking in the model, the growth rate $r_v$ is allowed to vary slightly in space. This would, for example, mimic microclimates, topography, or local weather heterogeneity

throughout a spatial area. To accomplish this, we add a small amount of variability to the annual seasonal period and amplitude of $r_v$ across the spatial grid as:

$$r_v(x,y,t) = r_{v,0} + A_{r_v}(x,y) \sin\left(\frac{2\pi t}{T_{\text{year}}} + \phi(x,y)\right) \quad (8)$$

We use $dt = 0.1$, saving only the last step of the day; this allows us to run our model at a higher temporal frequency while only recovering the daily time steps. We additionally use a small spin-up time at the beginning of the model run (maintaining $E = E_0$) to minimize inconsistencies due to numerical integration. The data from this model is used in Fig. 3 and Supplementary Fig. S6.

For the spatially extended Klausmeier pattern-formation model, we modify the approach of Bastiaansen et al.[46], keeping the classic two-equation setup introduced by Klausmeier[48]:

$$\frac{\partial N}{\partial t} = WN^2 - mN + D_n \nabla^2 N + \sigma \eta_N(\mathbf{x}, t) \quad (9)$$

$$\frac{\partial W}{\partial t} = a(t) - W - WN^2 + e\nabla^2 W \quad (10)$$

for vegetation biomass $N$, water $W$, vegetation mortality $m$, time-varying rainfall $a(t)$, Gaussian noise $\sigma$ and diffusion terms $D_n$ and $e$ for vegetation and water. The time-varying rainfall has an additional seasonal component with periodicity $T$ fixed as 365 days and amplitude $A$:

$$a(t) = a_{\text{base}}(t) + A \sin\left(\frac{2\pi t}{T_{\text{season}}}\right) \quad (11)$$

As rainfall declines following a logistic curve throughout the model run, vegetation patterns form and eventually collapse. We initialize the model at a high-biomass setting, and allow for a short spin-up time to stabilize the output. We use a time step $dt = 0.01$, capturing only the last step of the day for further analysis. The Klausmeier model is used for the analysis in Supplementary Fig. S7. Exact parameter settings for both spatial models can be found in the Supplement.

## Empirical data

**Glacier data.** Recent work has vastly expanded the amount and density of glacier velocity data available[31,32]. We use ITS_LIVE (version 2) velocity data sampled point-wise over the Landsat 8 era onward for two test glaciers: one in Alaska (RGI2000-v7.0-G-01-13271, [−140.847, 60.089]) and one in the Karakoram (RGI2000-v7.0-G-13-05693, [71.907, 38.837]) (Fig. 2)[40]. These glaciers were chosen because they show two different modes of destabilization—a multi-annual period of anomalously high velocity and a single surge period of extremely high velocity.

For each chosen glacier, we first select one single point in the center of the glacier tongue and extract all available velocity measurements. Each measurement represents the average velocity over a defined period; we first assign each velocity measurement to the midpoint of those periods. To minimize the confounding influence on CSD estimation from different numbers of measurements through time[28], we use a consensus-building approach. In short, we first create a set of equally-spaced annual points (4-day) and assign all velocity measurements to one of those equally-spaced bins. We then take $n = 100$ random iterations, choosing an equal number of points per bin based on the minimum number of contributing points across all bins. For example, if all equally-spaced bins have at least 5 points, we generate $n$ versions of our equally-spaced time series using a random sample of 5 points per time bin. We then average over the $n$ time series to create a single consensus time series of glacier velocity. In many cases, this is not significantly different from a rolling average; however, it explicitly accounts for the varying number of points per bin, so minimizes biases to the variance of the resulting time series.

In a second step, we generate a set of equally-spaced points along the glacier centerline at 500 m intervals. We further remove points in the upper portion of the glacier based on an underlying elevation model (Copernicus, 30 m)[81], with the assumption that the majority of the dynamics of interest are captured best in the lower reaches of each glacier. Since each point of the glacier is not guaranteed to have the same number of samples through time, we regularize the set of sampled points to an evenly-spaced time grid (4 days), leaving no-data values for missing points. This array of $n$ sample points through $t$ time steps is used directly with DMD without further pre-processing for the analysis shown in Fig. 4; the spatial pattern of velocities and missing data can be seen in Supplementary Fig. S16. We note that we choose a small temporal window (4 days) with the goal of maintaining some of the noise rather than over-smoothing; noise local to a single time series is de-emphasized during the DMD processing, and thus we err on the side of keeping too much noise rather than removing too much, as can happen during, for example, typical deseasoning procedures[21]. We confirm our results with a range of time samplings from 4 to 30 days (Supplementary Fig. S9); in each case the same critical transition can be seen before surge initiation.

**Vegetation data.** We use monthly-averaged Ku-Band Vegetation Optical Depth (VOD) data[50] covering the broader Amazon basin over the period 1987–2017 (0.25 dd spatial resolution). VOD data has been shown to be superior to optically-based vegetation indices for CSD analysis[4]. We choose monthly-averaged rather than daily-scale data for ease of processing; our rolling-window approach is computationally expensive for large spatio-temporal data sets. Previous work has shown that instrument changes in VOD data due to multiple overlapping satellite data sources can induce biases in stability estimates[28]; in short, changing signal-to-noise ratios between overlapping satellite data sets leave traces in both AC1 and variance that could be misinterpreted as stability changes.

Our approach attempts to reconstruct the underlying attractor despite noise; DMD is well-suited to this task even under variable noise levels. There is no evidence of a trend, and hence a bias, in the eigenvalue magnitudes as instruments are added or removed (Supplementary Fig. S2), despite clear biases in AC1 and variance. If noise levels are reasonable (less than the magnitude of the signal), we still capture the CSD eigenvalue cleanly despite changing noise levels (Supplementary Figs. S3, S5). We thus posit that VOD data—which has relatively high signal-to-noise ratios despite data mixing[28]—is suitable for analysis with our method.

**Amazon stability context data.** To compare our discovered patterns of Amazon stability loss to potential mechanistic explanations, we use forest cover change (version 1.12, 2000–2024, summed on a 0.25 dd grid)[51], Human Footprint Index (2022)[52], MODIS annual fire frequency (2001–2024)[53], annual precipitation sums and annual-scale sum changes based on CHIRPS data[54], and SPEI drought index (12-month scale) annual medians and change in annual median[55]. The data was all processed via Google Earth Engine[82]; the spatial maps can be seen in Supplementary Fig. S10.

**Greenland melt data.** We test our analysis on non-seasonal data that has been shown via other methods to exhibit signs of CSD[6]. This data is sourced from Trusel et al.[83], and is comprised of central west Greenland stack melt estimates from ice core records annually sampled since 1675[84]. We do not perform any other pre-processing on the data—we simply apply a rolling window ($t = 100$ years) and estimate changes in the largest eigenvalue using both S-Map and DMD (Supplementary Fig. S12).

## Data availability

All data is publicly available. Vegetation data can be found at https://doi.org/10.5281/zenodo.2575599[50]. Greenland melt data can be found via https://www.nature.com/articles/s41586-018-0752-4#Sec14[83]. ITS_LIVE glacier velocity data was accessed via the public Python API[31,32]. Forest cover change data is available at https://storage.googleapis.com/earthenginepartners-hansen/GFC-2024-v1.12/download.html[51]. MODIS fire frequency[53], CHIRPS precipitation data[54], and SPEI drought indices[55] are all available via Google Earth Engine[82]. Human Footprint Index data[52] are available here: https://doi.org/10.6084/m9.figshare.16571064.

## Code availability

Synthetic data creation, glacier velocity data access, and analysis scripts are publicly available on Zenodo[79].

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

## Acknowledgements

T.S. acknowledges support from the DFG STRIVE project (SM 710/2-1) and the Universität Potsdam Remote Sensing Computational Cluster. N.B. acknowledges funding by the Volkswagen Foundation. This is ClimTip contribution 132; the ClimTip project has received funding from the European Union's Horizon Europe research and innovation programme under grant agreement No. 101137601 (N.B.). This study received additional support from the European Space Agency Climate Change Initiative (ESA-CCI) Tipping Elements SIRENE project (contract no. 4000146954/24/I-LR) (N.B.). We thank Lana Blaschke for valuable discussion of the pros and cons of different spatial early warning signals.

## Author contributions

T.S. conceived and designed the study, processed the data, and performed the numerical analysis. T.S. wrote the manuscript with contributions from A.M., B.B., and N.B.

## Funding

## Competing interests

The authors declare no competing interests.
