## [Transparent Peer Review file · Nature Communications]

Predicting Instabilities in Transient Landforms and Interconnected Ecosystems

Corresponding Author: Dr Taylor Smith

Version 0:

Reviewer comments:

Reviewer #1

(Remarks to the Author)

The paper presents a novel and robust method for detecting early warning signals of critical transitions in complex systems. By combining concepts from Floquet theory and Dynamic Mode Decomposition (DMD), the authors develop a method to extract dominant eigenvalues directly from seasonally forced or nonstationary data—without requiring detrending or deseasoning. This is an important step forward in the study of tipping points, where conventional indicators often fail under periodic forcing or transient dynamics.

The authors demonstrate their approach on synthetic data and two high-impact case studies: glacier surging and vegetation dynamics in the Amazon. The method appears robust and flexible, and the application to real satellite-based time series is both timely and technically impressive. Overall, this work is a valuable contribution to the field of complex systems analysis and environmental forecasting. The paper is promising and well suited for publication in Nature Communications, but from my point of view two issues should be addressed which could help to enhance clarity and interpretability:

1. Since the method promises to overcome the backdraws of standard early warning indicators or other approaches, especially when detrending is needed (e.g. seasonal influences), it should be at least for one example quantitatively compared to one of these (detrended) indicators.
2. Especially in the Amazon case, the spatial patterns and eigenmodes are not clearly linked to known ecological processes (e.g., drought, fire, deforestation). More interpretative discussion or correlation with external data would improve the scientific value.

(Remarks on code availability)

Reviewer #2

(Remarks to the Author)

In this manuscript the authors present a method to measure critical slowing down in periodically forced systems and suggest that it works better than other methods. The results look indeed promising, though it could be that the AR and variance methods could perform better, for instance if the resolution was optimized or other detrending methods were used. The high AR1 values in figure 1 suggest that the resolution was too high.

My main point is that the method section is very technical. For me (I am not a mathematician) I could not understand what is happening due to the many unexplained mathematical terms and abbreviations. Examples are: Dynamic Mode Decomposition (DMD), Floquet Multipliers, Koopman operator, Monodromy matrix, False Nearest Neighbor (FNN), singular value decomposition (SVD), Moore Penrose pseudo inverse of matrix X, average mutual information, Ku-Band VOD data and STL processed. For a journal with a broad readership like Nature Communications, I think these terms should be better explained for non-mathematicians or avoided if possible. Also the “implementation details” gives little details about the implementation: for instance “choice of tau and m via Floquet multipliers” (how?) or “rule of thumb approximation of the embedding dimensions m based on Takens” (how?). Based on this description, I could not reproduce the work. As a reader I would at least want to know why all these techniques are needed and have an idea what they do. Because of this, I cannot well judge if the methodology are sound.

Another major point is that the manuscript does not give a way to determine whether the change is statistically significant. For instance AR1 methods use a null model to find whether the indicator is significantly increasing.

A minor point is that the described method is designed for periodically forced data, while this is not clear from the title or the abstract. You need to know the forcing period T .

In various figures with double axes we need to guess which is AR1 and which is variance.

(Remarks on code availability)

A link to the analyses scripts was not provided in this version.

Reviewer #3

(Remarks to the Author)

Smith et al., introduce a new method to assess the stability of different Earth systems based on eigenvalue tracking that can be applied to different types of remote sensing data. The manuscript follows a very logical structure, first presenting the application of the method to synthetic data and then moving to temporal and spatiotemporal data. I found the results very interesting and think that the approach has a lot of potential to help us better understand regime shifts across systems and scales. I have, however, some comments that I find important to address:

- The Introduction mentions a series of methods to measure early warning signals, going from the more traditional (and common) CSD indicators to more recent ones based on deep learning techniques (lines 23-25). However, the manuscript only compares the performance of dynamical mode decomposition with that of AC1 and variance. How does this new approach outperform more modern ways of measuring CSD?
- It would be very helpful to see an application of this approach to synthetic spatiotemporal data, similarly to what the authors do with temporal data. Spatially extended systems often show transitions between different patterned configurations as environmental conditions worsen (e.g., dryland vegetation [1]), but these transitions between pattern states feature very long transients with metastable configurations in which different pattern shapes may coexist. These transients make typical CSD indicators, even after detrending, unreliable early warning signals [2]. A deeper analysis of this approach on spatiotemporal synthetic data would be important to understand how the new method handles this and other issues intrinsic to spatially extended dynamics.
- The methods section does not provide enough details to understand the setup the authors used in the simulation example. For example, I had to go to the scripts to see how the model control parameter is made time-dependent. Having this information in the manuscript is important to understand what is behind the synthetic data and to understand the results better.

MINOR POINTS

- Rietkerk et al., Science 2021 [1] is a better reference than 55 and 56 to support the possible application of this approach to vegetation patterns
- In the same line as my previous comment regarding lack of method details, it would be helpful to define before the method what 'filtered maximum eigenvalue means'.

1 Rietkerk, M., Bastiaansen, R., Banerjee, S., van de Koppel, J., Baudena, M., & Doelman, A. (2021). Evasion of tipping in complex systems through spatial pattern formation. *Science*, 374(6564), eabj0359.

2 Veldhuis, M. P., Martinez-Garcia, R., Deblauwe, V., & Dakos, V. (2022). Remotely-sensed slowing down in spatially patterned dryland ecosystems. *Ecography*, 2022(10), e06139.

(Remarks on code availability)

I have not thoroughly reviewed the code because Python is not my main programming language. I did scan it, especially to find details about some model implementation that I could not find in the manuscript.

Version 1:

Reviewer comments:

Reviewer #1

(Remarks to the Author)

Since the authors addressed all my concerns in a meaningful manner, I recommend the article for publication without further changes.

(Remarks on code availability)

Reviewer #2

(Remarks to the Author)

The authors did a good job in this revision. All of my points were carefully considered. I found the paper however still hard to follow. There was indeed added a plain language part to the methods, which will help the reader for the overview, but it does not contain enough details to get an intuitive understanding why this method would work.

I appreciate the effort the authors made to show that it is difficult to use a bootstrap for the significance. It seems a disadvantage of this method that this seems impossible.

(Remarks on code availability)

Reviewer #3

(Remarks to the Author)

I thank the authors for their hard work in preparing this revised manuscript. They have fully addressed all my comments.

(Remarks on code availability)

I have gone through the code, but I am not qualified to review it in depth.

Reviewer 1

The paper presents a novel and robust method for detecting early warning signals of critical transitions in complex systems. By combining concepts from Floquet theory and Dynamic Mode Decomposition (DMD), the authors develop a method to extract dominant eigenvalues directly from seasonally forced or nonstationary data—without requiring detrending or deseasoning. This is an important step forward in the study of tipping points, where conventional indicators often fail under periodic forcing or transient dynamics.

The authors demonstrate their approach on synthetic data and two high-impact case studies: glacier surging and vegetation dynamics in the Amazon. The method appears robust and flexible, and the application to real satellite-based time series is both timely and technically impressive. Overall, this work is a valuable contribution to the field of complex systems analysis and environmental forecasting. The paper is promising and well suited for publication in Nature Communications, but from my point of view two issues should be addressed which could help to enhance clarity and interpretability:

Thank you for your positive evaluation of our work and your helpful comments. We have replied to both of them in-line below.

1. Since the method promises to overcome the backdraws of standard early warning indicators or other approaches, especially when detrending is needed (e.g. seasonal influences), it should be at least for one example quantitatively compared to one of these (detrended) indicators.

Thank you for this comment. We focused in the first version of the MS rather on showing where seasonal data failed (e.g., Figure 1b), but did not get into detrending and deseasoning the data, as there exist a wide range of methods to accomplish this task with different strengths and weaknesses. Indeed, not having to deseason/detrend the data is a key advantage of our method over current approaches – especially for spatio-temporal data. In our revision, we have modified Figure 1 to compare the restoring rate λ (Morr et al., 2024) on the raw seasonal data and λ on detrended and deseasoned data instead of showing AC1 or variance. We show a comparison of λ and AC1 for completeness in Figure 7 of this Reply; they generally have a similar character for deseasoned/detrended data. The updated Figure 1b of the manuscript – λ on deseasoned/detrended synthetic data – clearly shows differences in the implied system stability due to the choice of deseasoning/detrending method.

While the deseasoning/detrending approach can work well for systems with stable seasonality, it does not perform well in the case when seasonality varies significantly (e.g., in the glacier examples we show here), and partly motivates our work to find ways to assess stability directly on seasonal data. We have added a new Figure to the Supplement (Supplemental Figure S1) illustrating how common deseasoning/detrending methods can introduce biases, despite providing a nominally trend- and seasonality-free residual for use in the estimation of resilience. This Figure can also be seen in Figure 1 of this Reply.

Figure 1 – STL Deseasoning (blue) and simple mean-based deseasoning/detrending (orange) on synthetic data from Figure 1 (up to the transition point). There is significant seasonality left in the STL trend term, and far more variability in the STL seasonal amplitude than in the real (fixed) seasonal amplitude of the data. Both methods are designed to produce a residual without seasonal or long-term trend influences, with different underlying assumptions about the stability of the trend and seasonality.

2. Especially in the Amazon case, the spatial patterns and eigenmodes are not clearly linked to known ecological processes (e.g., drought, fire, deforestation). More interpretative discussion or correlation with external data would improve the scientific value.

Thank you for this comment. Indeed, it is very difficult to interpret the resulting spatio-temporal eigenvalue changes in terms of discrete physical processes, especially in a system as complex as the Amazon. We have examined several potential drivers as first-order comparisons (Figure 2 of this Reply), and find that no single one can be definitively matched to the eigenvalue mode changes we find (Figure

5 of the MS). While the spatial pattern of forest cover loss in the Amazon over the past few decades and a general Human Footprint Index (Mu et al., 2022) are suggestive (Figure 2b,c), the one-to-one coherence of the data sets is low; this is to some degree expected, as the spatial footprint of human influence can be much larger than what is directly seen on satellite data (Boulton et al., 2022). MODIS Burned Area maps also show relatively higher magnitude in the main southwestern Amazon region where the eigenvalue mode has changed the most (Figure 2a), but again do not match cleanly to the spatial pattern of our data.

Previous studies on resilience changes in the Amazon (e.g., Blashke et al. 2024, Boulton et al. 2022) have linked loss of resilience to regions with high human influence, low annual precipitation, and a high dependence on recycled moisture. Indeed, Blashke et al. (2024) found a similar spatial pattern of resilience changes (concentration of resilience loss in the south-west Amazon) with different data and methodology.

While an in-depth analysis of the direct link between these different drivers and our results would be interesting, it is beyond the scope of our current manuscript. We have added additional discussion of these possible drivers and the difficulties inherent in interpreting complex eigenmodes from our method to the Discussion section of our updated MS. Figure 2 of this Reply has also been added to the Supplement as Figure S10.

Figure 2 - Comparison of (A) CSD-Mode Magnitude Difference (cf. MS Figure 5) with changes in (B) Forest Cover (2000-2024, summed on a 0.25 dd grid) (Hansen et al., 2015), (C) Human Footprint Index (2022, Mu et al., 2022), (D) MODIS Fire Frequency (2001-2024, Giglio et al., 2021), (E) Annual Precipitation Sum and (F) Sum Change (CHIRPS, Funk et al. 2015), and (G) SPEI Drought Index 12-month Median and (H) Median Change (Vicente-Serrano et al., 2010). While the largest positive mode change (i.e., loss of stability, light blue areas, A) seems to correlate with forest cover loss, it is unlikely that this is the only driving factor.

References

- Blaschke, L. L.; Nian, D.; Bathiany, S.; Ben-Yami, M.; Smith, T.; Boulton, C. A. & Boers, N. Spatial Correlation Increase in Single-Sensor Satellite Data Reveals Loss of Amazon Rainforest Resilience. *Earth's Future*, **2024**, *12*, e2023EF004040
- Boulton, C. A.; Lenton, T. M. & Boers, N. Pronounced loss of Amazon rainforest resilience since the early 2000s. *Nature Climate Change*, *Nature Publishing Group*, **2022**, *12*, 271-278. <https://doi.org/10.1038/s41558-022-01287-8>
- Funk, C., Peterson, P., Landsfeld, M. *et al.* The climate hazards infrared precipitation with stations—a new environmental record for monitoring extremes. *Sci Data* **2**, 150066 (2015). <https://doi.org/10.1038/sdata.2015.66>
- Giglio, L.; Justice, C.; Boschetti, L. & Roy, D. MODIS/Terra+Aqua Burned Area Monthly L3 Global 500m SIN Grid V061 [Data set] *NASA Land Processes Distributed Active Archive Center*, **2021** <https://doi.org/10.5067/MODIS/MCD64A1.061>
- Hansen, M. C., P. V. Potapov, R. Moore, M. Hancher, S. A. Turubanova, A. Tyukavina, D. Thau, S. V. Stehman, S. J. Goetz, T. R. Loveland, A. Kommareddy, A. Egorov, L. Chini, C. O. Justice, and J. R. G. Townshend. 2013. High-Resolution Global Maps of 21st-Century Forest Cover Change. *Science* 342 (15 November): 850-53. Data available on-line from: <https://glad.earthengine.app/view/global-forest-change>
- Morr, A. & Boers, N. Detection of Approaching Critical Transitions in Natural Systems Driven by Red Noise. *Phys. Rev. X* **14**, (2024). <https://doi.org/10.1103/PhysRevX.14.021037>
- Mu, H., Li, X., Wen, Y. *et al.* A global record of annual terrestrial Human Footprint dataset from 2000 to 2018. *Sci Data* **9**, 176 (2022). <https://doi.org/10.1038/s41597-022-01284-8>
- Vicente-Serrano, S. M.; Beguería, S. & López-Moreno, J. I. A Multiscalar Drought Index Sensitive to Global Warming: The Standardized Precipitation Evapotranspiration Index. *Journal of Climate*, *American Meteorological Society*, **2010**, *23*, 1696 - 1718. 10.1175/2009JCLI2909.1

Reviewer 2

In this manuscript the authors present a method to measure critical slowing down in periodically forced systems and suggest that it works better than other methods. The results look indeed promising, though it could be that the AR and variance methods could perform better, for instance if the resolution was optimized or other detrending methods were used. The high AR1 values in figure 1 suggest that the resolution was too high.

Thank you for your comments on the manuscript. Indeed, there are several ways to optimize the detrending and deseasoning of seasonally-forced data; however, each of them has potential drawbacks and requires assumptions to be made about how to partition the system into seasonal, trend, and residual components (e.g., Figure 1 of this Reply). For example, the common approach of removing a long-term monthly climatology implies that the seasonal cycle has not changed at all through the whole set of study years. Another common method – seasonal trend decomposition via Loess (STL) – requires a few parameters related to the assumed smoothness and shape of the seasonality to be set. While both methods work, they potentially introduce biases into the resulting stability estimates. This can clearly be seen in applying two detrending/deseasoning methods on the synthetic data (Figure 1 of this Reply, new Supplemental Figure S1). These differences are also carried forward into the estimated AC1, variance, and restoring rate λ on those deseasoned/detrended time series (Figure 3b of this Reply). We have replaced AC1 with the more noise-robust λ (Morr et al., 2024) after the comments of another reviewer; these metrics are comparable CSD indicators. A comparison of λ and AC1 can be seen in Figure 7 of this Reply.

If we resample the raw and deseasoned data before calculating λ (or AC1), it is relatively lower and shows a subtle rise towards the transition point; this change, however, is always to some degree biased by the deseasoning procedure. As a second test, we have generated an additional tipping model, which has lower inherent autocorrelation and a more extreme increase in λ as a precursor. In this model, the λ starts lower and rises quickly – especially for the version of the model run without seasonality (Figure 3b, black dashed line). The Floquet multipliers again provide an early warning signal of the oncoming transition in this model.

Figure 3 – Additional synthetic model, which also shows clearer precursor signals due to lower inherent autocorrelation in the model. Based on a logistic rather than pitchfork tipping model. Restoring rate (λ) used on (B) instead of AC1, as this is more robust to changing noise characteristics (Morr et al., 2024). Clipping is apparent in λ and DMD eigenvalues due to the imposed lack of negative values in the model. A full description of the model can be found in the updated Methods section, and this new Figure can be seen as Supplemental Figure S15.

Finally, we have also generated an additional version of the pitchfork bifurcation model used in Figure 1 that uses a linear ramp for the forcing rather than a sigmoidal (tanh) forcing. This again shows the clear capture of an early warning signal by our method (updated Figure 1 of the MS) for both slow forcing change (Figure 4 of this Reply, Figure 1 of the updated MS) and fast forcing change (Figure 5 of this Reply, Supplemental Figure S14).

Figure 4 – Pitchfork bifurcation model with slow ramp forcing. Clear precursor signals are seen in the deseasoned/detrended data due to the longer ramp over the transition. Based on the same pitchfork tipping model as the original manuscript Figure 1, but with linear rather than sigmoidal forcing. This new Figure replaces Figure 1 of the MS, Figure 1 of the MS has been moved to the Supplement for completeness. Clearer early warning from the DMD-based approach due to the slow ramp forcing that is easy to separate from the noise and seasonal terms.

Figure 5 - Pitchfork bifurcation model with fast forcing change (tanh, sigmoidal). Same as Figure 4 of this Reply except for the speed of the forcing over the transition. Clear bias in STL-deseasoned data which completely misses the increase in λ before the transition.

My main point is that the method section is very technical. For me (I am not a mathematician) I could not understand what is happening due to the many unexplained mathematical terms and abbreviations. Examples are: Dynamic Mode Decomposition (DMD), Floquet Multipliers, Koopman operator, Monodromy matrix, False Nearest Neighbor (FNN), singular value decomposition (SVD), Moore Penrose pseudo inverse of matrix X , average mutual information, Ku-Band VOD data and STL processed. For a journal with a broad readership like Nature Communications, I think these terms should be better explained for non-mathematicians or avoided if possible. Also the “implementation details” gives little details about the implementation: for instance “choice of tau and m via Floquet multipliers” (how?) or “rule of thumb approximation of the embedding dimensions m based on Takens” (how?). Based on this description, I could not reproduce the work. As a reader I would at least want to know why all these techniques are needed and have an idea what they do. Because of this, I cannot well judge if the methodology are sound.

Thank you for this comment. We included such a rigorous treatment of the mathematical background to properly justify our novel method and situate our derived eigenvalue magnitudes with respect to the commonly-used stability estimates present in many other papers. We agree, however, that this became rather involved. In our revision, we have re-organized the methods section to concentrate the

mathematical and technical details into their own section, and provide a more ‘plain language’ methodological description as a separate first part of the Methods. We hope that this will improve the legibility of the MS for a broad audience, as well as still provide the mathematical justification for our work for those who are interested. Reviewer 3, for example, asked for additional mathematical details on the synthetic models to be included in the Methods section.

Another major point is that the manuscript does not give a way to determine whether the change is statistically significant. For instance AR1 methods use a null model to find whether the indicator is significantly increasing.

Thank you for raising this point, as this aspect of our proposed method was not stated clearly enough in our previous version of the manuscript. There is a fundamental difference in how the results of our stability estimates are interpreted with respect to anticipating transitions as opposed to other conventional indicators of CSD (e.g., AC1, variance, λ). AC1, variance, and λ are typically assessed in rolling windows, allowing for the analysis of trends that can be used as an “early warning” of an impending transition when their trends turn significant. In our method, however, the estimated eigenvalues of embedded dynamics express an explicit destabilisation at the point where the maximal (non-seasonal) eigenvalue crosses 1 from below. This threshold can be interpreted as an indicator of stability loss and is, in some sense, conservative, as trends in eigenvalues or CSD indicators more generally can also emerge from other changes in the system unrelated to destabilisation (Rietkerk et al., 2025). It would, in principle, be possible to extend our method, however, to assess oncoming destabilization based on when the maximum non-seasonal eigenvalue starts to consistently approach 1 from below, providing the possibility of warning far before the stability threshold of 1 is crossed. In such a case, we would indeed need to assess the statistical significance of that increase towards 1.

Your comment raises an interesting question: If we wanted to perform the kind of trend analysis that is done with CSD indicators in other resilience publications, how could we check for significance in our eigenvalue changes? One common approach for testing the significance of changes in, e.g., AC1 on a time series is to create phase-shuffled surrogates and test how many of them also show increasing (or decreasing) trends in AC1. This works because the phase shuffling leaves the underlying autocorrelation and linear features of the data untouched, while destroying non-linear parts. If the changes in AC1 are attributable to a non-linear process, the phase shuffled surrogates should not have a trend in autocorrelation, as they are (by construction) linear; using the confidence bounds of the phase surrogates allows us to either reject or accept the null model of a simple linear process.

This approach is, however, not directly mappable to our eigenvalue-based approach, as we perform pointwise decompositions in embedded space – phase shuffling the time series destroys the embedded geometry of the data (as well as the seasonal-scale phase), and can yield wildly different results with our method. In short, the phase is intrinsic to how the underlying attractor is defined in embedded space, and randomizing those phases destroys the attractor we want to find. This hence rules out the estimation of eigenvalues of that attractor on phase-shuffled surrogates.

An alternative null model can be constructed by temporally-shuffling the time series, with the goal of preserving local-scale features needed for DMD decomposition while destroying long-term structure in the time series (Dudek et al., 2013; Polantis and Romano, 1994; McKinnon et al., 2016). This would test

whether any resulting trend (e.g., Kendall-Tau on the maximum non-seasonal eigenvalue) is attributable to random variation or to an actual increase in eigenvalue magnitude. To maintain seasonality and the embedded structure required to find the attractor, we need to impose the condition that blocks are drawn starting from the same day of the year: if, for example, a block ends on May 31st, the following block must start on June 1st (with a small amount of leeway, e.g., Feb 29th). This maintains seasonality while shuffling the ordering, and also maintains embedded states as coherent blocks. In this way, we can construct a set of n surrogates and test how many of them show increases (or decreases) in the Kendall-Tau of the dominant non-seasonal eigenvalue.

The drawback of this method is inconsistencies at the block margins, which can be quantified with a 'contamination fraction' measuring the percentage of embedded DMD windows that will be influenced by block margins. If we use small blocks, there are many such inconsistencies within each DMD estimation window. If we use large (e.g., 1+ year) blocks, we have a limited number of blocks to shuffle to create our surrogates, and hence lose statistical power. As the impact of these margin inconsistencies on DMD eigenvalue magnitudes and partitioning (e.g., between the seasonal and noise eigenvectors) is not clear, it is hard to choose an ideal block size and surrogate methodology. It is possible that the block margins are simply adding 'noise' in the DMD estimation, generally get partitioned into the low-magnitude eigenvalues, and have a minimal impact on the assessed trends in the dominant eigenvalue. It is also possible, however, that the block margins have strong non-linear impacts on the DMD estimates, with unknown bias potential on the strength and positioning of the CSD-eigenvalue of interest.

We tested three methods of generating surrogates: (1) a Generalized Block Bootstrap (Dudek et al. 2013), (2) a Stationary Bootstrap (Polantis and Romano, 1994) (both using a range of block sizes), and (3) a pure annual-cycle bootstrap (McKinnon et al., 2016). Using $n=1000$ surrogates, we find that a block size of 270 days leaves us with ~25% of the DMD windows contaminated by margin effects, and yields a p -value of 0.07 for the observed increasing trend of the maximum non-seasonal eigenvalue shown in Figure 5 of this Reply (previous manuscript Figure 1); the pure-cycle bootstrap shows a similar p -value for a slightly smaller contamination fraction. Smaller block sizes with more contamination decrease the p -value of the trend due to the increased randomness in the surrogates; larger block sizes reduce the p -value to ~0.1 due to a decrease in the randomness of the surrogates. A comparison of the impacts of block sizes and surrogate methods can be seen in Figure 6 of this Reply.

Figure 6 - Exploration of p -values and contamination fractions for different surrogate generation methods ($n=1000$). (A) Comparison of p -values and block length for a purely seasonal bootstrap (Dudek et al., 2013), a stationary bootstrap (Polantis and Romano, 1994), and a cyclic bootstrap (McKinnon et al., 2016). (B) Same for comparison of contamination fraction and block length. (C) Comparison of p -values and contamination fractions. (D) Histogram comparison for ‘best’ block length for the seasonal and stationary bootstraps with the cyclic bootstrap. Kendall-Tau of the original series plotted as a vertical line; p -values reported for each method in the legend.

We hesitate to conclude, however, that any of these methods are ideal means of assessing eigenvalue trend significance. There is limited research on bootstrap methods in embedded state-space, and a rigorous analysis of which surrogate-generation methods are mathematically correct for our novel DMD eigenvalue estimation method is outside of the scope of our study. We further note that while the straightforward bootstraps we present here for the 1D case are likely appropriate, it is not clear how such surrogates would map to the spatial or multi-dimensional case. For example, one could either opt to shuffle our spatial fields only in time or to also shuffle their spatial components. It is also unclear how

large the temporal block size needs to be to properly preserve the embedded state-space in a spatial field versus in a 1D system.

As the generation of surrogates that preserve spatio-temporal structure – particularly with regard to state-space embedding and the particularities of eigenvalue estimation with DMD – has not been explored in the literature, we will not present trends in the dominant non-seasonal eigenvalue in this work, as we cannot concisely provide trend significance in a mathematically rigorous manner. The work we present here is not geared towards assessing sustained increases in eigenvalue magnitude through time; this, however, would be a great candidate for future research. As eigenvalues can oscillate quite strongly through time, defining trends outside of the near-transition-zone remains difficult, and assessing their statistical significance in a rigorous way remains an unexplored area of research.

A minor point is that the described method is designed for periodically forced data, while this is not clear from the title or the abstract. You need to know the forcing period T .

Thank you for this comment. While one of the clear advantages of our method is that it works natively on periodic systems, it is not only useful in that case – the eigenvalue-based early-warning signals are also useful on non-periodic data (e.g., Supplemental Figures S11, S12), and are themselves an extension of recently proposed methods using S-Map to estimate eigenvalues (Grziwotz et al., 2023). We have added a note that our method works for periodically-forced data of a known periodicity to the abstract, such as for annually-driven systems.

In various figures with double axes we need to guess which is AR1 and which is variance.

We have updated the Figures throughout the MS and Supplement to make this clearer using colored axes.

References

- Dudek, A. E.; Leśkow, J.; Paparoditis, E. & Politis, D. N. A generalized block bootstrap for seasonal time series. *Journal of Time Series Analysis, Wiley Online Library*, 2014, 35, 89-114 <https://doi.org/10.1002/jtsa.12053>
- Grziwotz, F. et al. Anticipating the occurrence and type of critical transitions. *Science Advances* 9, eabq4558 (2023). <https://www.science.org/doi/abs/10.1126/sciadv.abq4558>
- McKinnon, K., Rhines, A., Tingley, M. et al. Long-lead predictions of eastern United States hot days from Pacific sea surface temperatures. *Nature Geosci* 9, 389–394 (2016). <https://doi.org/10.1038/ngeo2687>
- Politis, D. N. & Romano, J. P. The stationary bootstrap. *Journal of the American Statistical association, Taylor & Francis*, 1994, 89, 1303-1313 <https://doi.org/10.1080/01621459.1994.10476870>
- Rietkerk, M., Skiba, V., Weinans, E., Hébert, R. & Laepple, T. Ambiguity of early warning signals for climate tipping points. *Nat. Clim. Chang.* 15, 479–488 (2025). <https://doi.org/10.1038/s41558-025-02328-8>

Reviewer 3

Smith et al., introduce a new method to assess the stability of different Earth systems based on eigenvalue tracking that can be applied to different types of remote sensing data. The manuscript follows a very logical structure, first presenting the application of the method to synthetic data and then moving to temporal and spatiotemporal data. I found the results very interesting and think that the approach has a lot of potential to help us better understand regime shifts across systems and scales. I have, however, some comments that I find important to address:

Thank you for your positive review of our work, and your helpful comments. We have addressed each of them in-line below.

The Introduction mentions a series of methods to measure early warning signals, going from the more traditional (and common) CSD indicators to more recent ones based on deep learning techniques (lines 23-25). However, the manuscript only compares the performance of dynamical mode decomposition with that of AC1 and variance. How does this new approach outperform more modern ways of measuring CSD?

Thank you for this comment. Most CSD methods face the same biases due to seasonality – autocorrelation, variance, spatial autocorrelation, flickering, and spectral properties are all influenced by periodic movement in a time series, which is one of the main strengths of our new method. To move beyond the most common (AC1 and variance) methods of estimating CSD, we have added a recently proposed means of estimating resilience that is robust with respect to changing noise characteristics (Morr et al., 2024) to our analysis of synthetic 1D models (λ); this method is also biased by seasonality, but performs similarly to AC1 on deseasoned/detrended or non-seasonal data (Figure 7 of this Reply), while being less influenced by underlying noise.

Figure 7 - Comparison of AC1 and λ (Morr et al., 2024) for the data presented in updated Figure 1 of the manuscript, across different means of deseasoning/detrending, as well as the raw seasonal and non-seasonal models. Both methods show similar characteristics for the 1D model.

In response to your other comments, we have added two synthetic spatial models to the manuscript, which we assess using spatial variance, spatial skewness, temporal autocorrelation, and Moran's I as the 'baseline'. We have included temporal autocorrelation and Moran's I as first-order comparisons for the real-world spatial cases. It is important to note, however, that interpreting, e.g., Moran's I on spatially disconnected systems is difficult. For example, in a synthetic system, it is possible to assume that all pixels are connected and can thus contribute to the changes in the spatial structure that could be used as early-warning signals. In the real world, however, there are many ways to disconnect (eco)systems, with unclear influence on the implied values of spatial autocorrelation, Moran's I, skewness, etc. Direct grid-to-grid temporal autocorrelation makes fewer assumptions about data connectedness, which likely makes it the most useful baseline comparison for our real-world spatial data; we include Moran's I here as well for completeness.

While it is possible that machine learning methods may provide an alternative means of estimating stability changes on seasonal data, this is not something that has – to our knowledge – been tested before. There also does not exist a machine learning framework that can be applied directly to our data. A crucial issue in this context would be the availability of suitable training data. In order to test such an approach, we would need to build, train, and validate a model on data with and without transitions; existing models to predict critical transitions would only work on the data and situations that they have been trained for, so it would remain questionable to apply such a machine learning model to observations. While we find this an interesting idea, it is beyond the scope of this paper and would be better suited to a further research project.

It would be very helpful to see an application of this approach to synthetic spatiotemporal data, similarly to what the authors do with temporal data. Spatially extended systems often show transitions between different patterned configurations as environmental conditions worsen (e.g., dryland vegetation [1]), but these transitions between pattern states feature very long transients with metastable configurations in which different pattern shapes may coexist. These transients make typical CSD indicators, even after detrending, unreliable early warning signals [2]. A deeper analysis of this approach on spatiotemporal synthetic data would be important to understand how the new method handles this and other issues intrinsic to spatially extended dynamics.

1 Rietkerk, M., Bastiaansen, R., Banerjee, S., van de Koppel, J., Baudena, M., & Doelman, A. (2021). Evasion of tipping in complex systems through spatial pattern formation. *Science*, 374(6564), eabj0359.

2 Veldhuis, M. P., Martinez-Garcia, R., Deblauwe, V., & Dakos, V. (2022). Remotely-sensed slowing down in spatially patterned dryland ecosystems. *Ecography*, 2022(10), e06139.

This is a great idea. We have implemented a simple vegetation model with diffusion (Dakos et al., 2010), modified to also include seasonality in the forcing and a spatially variable growth rate on each individual pixel. A description of the model and how we parameterized it is found in the updated Methods section, with a parameter table in the Supplement and code to implement the model in the Code Repository. We find a clear early-warning signal with our method, and that it doesn't have the seasonal biases which are apparent in previously proposed spatial early warning signals (Figure 8 of this Reply and Figure 3 of the updated MS).

Figure 8 – Spatial vegetation tipping model. (A) Slices from the spatial grid of the vegetation model. (B) Mean vegetation state, with vertical lines showing the location of spatial snapshots from (A). (C) Typical spatial early-warning signals on STL detrended/deseasoned data: spatial skewness (purple), temporal autocorrelation (red), spatial variance (blue), and Moran’s I (black). Temporal autocorrelation seems to be the least biased by seasonality. (D) Eigenvalue magnitudes, showing a crossing of the dominant non-seasonal eigenvalue well before the transition. The seasonal eigenvalue is slightly unstable (magnitude >1) due to the slow increase in seasonal amplitude in the model as the resilience of the system decreases, which drives the seasonal amplitude slightly higher through time.

Pattern-forming systems are a much trickier problem – as you note, quasi-stable patterns limit the application of typical early-warning signals. To test whether our method works in this case, we set up a simple pattern-forming model (Klausmeier, based on that presented in Bastiaanen et al., 2018), with additional seasonal oscillations (see Methods for discussion of the model, and Supplement for a full parameter table). We find that there are large increases in the Floquet multipliers when patterns first form, and again when patterns start to collapse (Figure 9 of this Reply). The difficulty is in the time scale and mechanism of the two transitions (full vegetation -> pattern -> desert). For the first transition, this is

not a real CSD event, but a spatial reorganization, which should not have the same increases in CSD-metrics as the second, saddle-node bifurcation from patterned vegetation to desert. Detecting these changes with typical spatial early-warning signals is difficult; from our first-order test, however, it seems like the Floquet DMD approach could detect the onset of pattern formation. It also seems to map when a system is in an unstable pattern-forming state (eigenvalue strongly above 1), as well as when the patterns stabilize before the collapse (eigenvalue crossing from above 1 as patterns change from linear and rapidly shifting to stable dots). Further work, however, would be needed to explore this approach more deeply and better differentiate the spatial reorganization and CSD signals. In short, DMD eigenvalues are also sensitive to spatial reorganization and not only CSD-like events; this expands their utility but also complicates their direct interpretation. We have added a Discussion of pattern formation to the manuscript and the additional model description to our Methods. Figure 9 of this Reply is also now in the Supplement (Supplemental Figure S7).

Figure 9 - Eigenvalue tracking on a simple pattern-formation model with seasonality. (A) Spatial maps of snapshots in time linked to different points in the time series. (B) Mean vegetation state, with vertical lines showing the location of spatial snapshots from (A). (C) Spatial early warning signals (on data detrended/deseasoned with STL) have different responses to the onset of patterns and the collapse of the system. (D) Eigenvalue tracking, showing a strong and slightly unstable seasonal eigenvalue as well as an increase in the dominant non-seasonal eigenvalue up until the first one-crossing point. There are also high and sustained eigenvalue magnitudes during the pattern formation period, followed by a decrease as patterns decay. Final short increase in eigenvalue magnitude above one before collapse (marked as last crossing in A,B).

The methods section does not provide enough details to understand the setup the authors used in the simulation example. For example, I had to go to the scripts to see how the model control parameter is made time-dependent. Having this information in the manuscript is important to understand what is behind the synthetic data and to understand the results better.

Thank you for this comment. In our updated manuscript, we have significantly reorganized the Methods to (1) make the model parameterization clearer, (2) describe the additional models, and (3) split the 'basic overview' of our method away from the mathematical framework to provide an easier introduction to the method for a general audience. We have fully described each model, and we also now provide a parameter table for each model in the Supplement to make this clearer.

MINOR POINTS

Rietkerk et al., Science 2021 [1] is a better reference than 55 and 56 to support the possible application of this approach to vegetation patterns.

We have updated this reference.

In the same line as my previous comment regarding lack of method details, it would be helpful to define before the method what 'filtered maximum eigenvalue means'.

We have added additional clarification of the terms in our reorganized Methods section, and put a short note in the main text defining the filtered non-seasonal eigenvalue before it is first used in a figure.

Remarks on code availability

I have not thoroughly reviewed the code because Python is not my main programming language. I did scan it, especially to find details about some model implementation that I could not find in the manuscript.

Thank you for this comment. We have updated the Methods section to make the full parameterization of our models clearer, and added additional in-code comments to help those who want to apply our methods. We have also added parameter tables in the Supplementary materials.

References

Bastiaansen, Robbin, et al. "Multistability of model and real dryland ecosystems through spatial self-organization." *Proceedings of the National Academy of Sciences* 115.44 (2018): 11256-11261.

<https://doi.org/10.1073/pnas.1804771115>

Dakos, V., van Nes, E.H., Donangelo, R. *et al.* Spatial correlation as leading indicator of catastrophic shifts. *Theor Ecol* **3**, 163–174 (2010). <https://doi.org/10.1007/s12080-009-0060-6>

Morr, A. & Boers, N. Detection of Approaching Critical Transitions in Natural Systems Driven by Red Noise. *Phys. Rev. X* **14**, (2024). <https://doi.org/10.1103/PhysRevX.14.021037>

Reply to Reviewers – Smith et al, Nature Communications

Reviewer #1:

Since the authors addressed all my concerns in a meaningful manner, I recommend the article for publication without further changes.

Thank you for your time and effort in reviewing the paper!

Reviewer #2:

The authors did a good job in this revision. All of my points were carefully considered.

Thank you for your helpful comments, and we are glad that we have been able to address your main concerns.

I found the paper however still hard to follow. There was indeed added a plain language part to the methods, which will help the reader for the overview, but it does not contain enough details to get an intuitive understanding why this method would work.

In this revision, we have added further introduction material to the start of the Methods section to give a general goal of the method. We have also expanded that first Methods section to add more intuitive and plain-language framing. We hope that this further improves the legibility of the paper.

I appreciate the effort the authors made to show that it is difficult to use a bootstrap for the significance. It seems a disadvantage of this method that this seems impossible.

We agree – the difficulty of building a null model is a key limitation of the method. We have added a short caveats section around this to the Discussion in order to motivate further work on developing new methods of error propagation and uncertainty estimation for the timing of system transitions.

Reviewer #3:

I thank the authors for their hard work in preparing this revised manuscript. They have fully addressed all my comments.

Thank you for your time and effort in reviewing the paper!